# FASTER GRADIENT METHODS FOR HIGHLY-SMOOTH STOCHASTIC BILEVEL OPTIMIZATION

**Lesi Chen and Junru Li**
Tsinghua University & Shanghai Qizhi Institute, China
{chenlc23,jr-li24}@mails.tsinghua.edu.cn

**El Mahdi Chayti**
École Polytechnique Fédérale de Lausanne (EPFL), Switzerland
el-mahdi.chayti@epfl.ch

**Jingzhao Zhang** [†]
Tsinghua University & Shanghai Qizhi Institute,China
jingzhaoz@mail.tsinghua.edu.cn

## ABSTRACT

This paper studies the complexity of finding an $\epsilon$-stationary point for stochastic bilevel optimization when the upper-level problem is nonconvex and the lower-level problem is strongly convex. Recent work proposed the first-order method, F$^2$SA, achieving the $\tilde{\mathcal{O}}(\epsilon^{-6})$ upper complexity bound for first-order smooth problems. This is slower than the optimal $\Omega(\epsilon^{-4})$ complexity lower bound in its single-level counterpart. In this work, we show that faster rates are achievable for higher-order smooth problems. We first reformulate F$^2$SA as approximating the hyper-gradient with a forward difference. Based on this observation, we propose a class of methods F$^2$SA-$p$ that uses $p$th-order finite difference for hyper-gradient approximation and improves the upper bound to $\tilde{\mathcal{O}}(p\epsilon^{-4-2/p})$ for $p$th-order smooth problems. Finally, we demonstrate that the $\Omega(\epsilon^{-4})$ lower bound also holds for stochastic bilevel problems when the high-order smoothness holds for the lower-level variable, indicating that the upper bound of F$^2$SA-$p$ is nearly optimal in the region $p = \Omega(\log \epsilon^{-1} / \log \log \epsilon^{-1})$.

## 1 INTRODUCTION

Many machine learning problems, such as meta-learning (Rajeswaran et al., 2019), hyper-parameter tuning (Bao et al., 2021; Franceschi et al., 2018; Mackay et al., 2019), adversarial training (Goodfellow et al., 2020), and reinforcement learning (Yang et al., 2019; Hong et al., 2023; Zeng et al., 2024; Zeng & Doan, 2024) can be abstracted as solving the following bilevel optimization problem:

$$\min_{\boldsymbol{x} \in \mathbb{R}^{d_x}} \varphi(\boldsymbol{x}) = f(\boldsymbol{x}, \boldsymbol{y}^*(\boldsymbol{x})), \quad \boldsymbol{y}^*(\boldsymbol{x}) = \arg \min_{\boldsymbol{y} \in \mathbb{R}^{d_y}} g(\boldsymbol{x}, \boldsymbol{y}), \tag{1}$$

We call $f$ and $g$ the upper-level and lower-level functions, respectively, and call $\varphi$ the *hyper-objective*. In this paper, we consider the most common *nonconvex-strongly-convex* setting where $f : \mathbb{R}^{d_x} \to \mathbb{R}$ is smooth and possibly nonconvex, and $g : \mathbb{R}^{d_y} \to \mathbb{R}$ is smooth jointly in $(\boldsymbol{x}, \boldsymbol{y})$ and strongly convex in $\boldsymbol{y}$. Under the lower-level strong convexity assumption, the implicit function theorem (Dontchev & Rockafellar, 2009) indicates the following closed form of the hyper-gradient

$$\nabla \varphi(\boldsymbol{x}) = \nabla_x f(\boldsymbol{x}, \boldsymbol{y}^*(\boldsymbol{x})) - \nabla_{xy}^2 g(\boldsymbol{x}, \boldsymbol{y}^*(\boldsymbol{x}))[\nabla_{yy}^2 g(\boldsymbol{x}, \boldsymbol{y}^*(\boldsymbol{x}))]^{-1} \nabla_y f(\boldsymbol{x}, \boldsymbol{y}^*(\boldsymbol{x})). \tag{2}$$

Following the works in nonconvex optimization (Carmon et al., 2020; 2021; Arjevani et al., 2023), we consider the task of finding an $\epsilon$-stationary point of $\varphi$, *i.e.*, a point $\boldsymbol{x} \in \mathbb{R}^{d_x}$ such that

---
[†] The corresponding author.

$\|\nabla\varphi(\boldsymbol{x})\| \leq \epsilon$. Motivated by many real machine learning tasks, we study the stochastic setting, where the algorithms only have access to the stochastic derivative estimators of both $f$ and $g$.

The first efficient algorithm BSA Ghadimi & Wang (2018) for solving the stochastic bilevel problem leverages both stochastic gradient and Hessian-vector-product (HVP) oracles to find an $\epsilon$-stationary point of $\varphi(\boldsymbol{x})$. Subsequently, Ji et al. (2021) proposed stocBiO by incorporating multiple enhanced designs in BSA to improve the complexity. Yang et al. (2023b) proposes FdeHBO that uses finite-differences to estimate HVP vectors. However, all these methods require the stochastic Hessian assumption (5) on the lower-level function, which is stronger than the standard SGD assumption.

Kwon et al. (2023) proposed the first fully first-order method F$^2$SA that works under standard SGD assumptions on both $f$ and $g$ (Assumption 2.1). The main idea is to solve the following penalty problem (Liu et al., 2022; 2023; Shen & Chen, 2023; Shen et al., 2025b; Lu & Mei, 2024; 2026):

$$\min_{\boldsymbol{x}\in\mathbb{R}^{d_x},\boldsymbol{y}\in\mathbb{R}^{d_y}} f(\boldsymbol{x},\boldsymbol{y}) + \lambda\left(g(\boldsymbol{x},\boldsymbol{y}) - \min_{\boldsymbol{z}\in\mathbb{R}^{d_y}} g(\boldsymbol{x},\boldsymbol{z})\right), \tag{3}$$

where $\lambda$ is taken to be sufficiently large such that $\lambda = \Omega(\epsilon^{-1})$. If we interpret $\lambda$ as the Lagrangian multiplier, then Problem (3) can be viewed as the Lagrangian function of the constrained optimization $\min_{\boldsymbol{x}\in\mathbb{R}^{d_x},\boldsymbol{y}\in\mathbb{R}^{d_y}} f(\boldsymbol{x},\boldsymbol{y}), \text{s.t.} g(\boldsymbol{x},\boldsymbol{y}) \leq g(\boldsymbol{x},\boldsymbol{y}^*(\boldsymbol{x}))$. Thanks to Danskin's theorem, the gradient of the Problem (3) only involves gradient information. Therefore, F$^2$SA does not require the stochastic Hessian assumptions (5). More importantly, by directly leveraging gradient oracles instead of more expensive HVP oracles, the F$^2$SA is more efficient in practice (Shen et al., 2025a; Xiao & Chen, 2025; Jiang et al., 2025) and is also the only method that can be scaled to 32B sized large language model (LLM) training (Pan et al., 2024).

Kwon et al. (2023) proved that the F$^2$SA method finds an $\epsilon$-stationary point of $\varphi(\boldsymbol{x})$ with $\tilde{\mathcal{O}}(\epsilon^{-3})$ first-order oracle calls in the deterministic case and $\tilde{\mathcal{O}}(\epsilon^{-7})$ stochastic first-order oracle (SFO) calls in the stochastic case. Recently, Chen et al. (2025b) showed the two-time-scale stepsize strategy improves the upper complexity bound of F$^2$SA method to $\tilde{\mathcal{O}}(\epsilon^{-2})$ in the deterministic case, which is optimal up to logarithmic factors. However, the direct extension of their method in the stochastic case leads to the $\tilde{\mathcal{O}}(\epsilon^{-6})$ SFO complexity (Chen et al., 2025b; Kwon et al., 2024a), which still has a significant gap between the $\Omega(\epsilon^{-4})$ lower bound for SGD (Arjevani et al., 2023). It remains open whether optimal rates for stochastic bilevel problems can be achieved for fully first-order methods.

In this work, we revisit F$^2$SA and interpret it as using forward difference to approximate the hyper-gradient. Our novel interpretation in turn leads to straightforward algorithm extensions for the F$^2$SA method. Observing that the forward difference used by F$^2$SA only has a first-order error guarantee, a natural idea to improve the error guarantee is to use higher-order finite difference methods. For instance, we know that the central difference has an improved second-order error guarantee. Based on this fact, we can derive the F$^2$SA-2 method that solves the following symmetric penalty problem:

$$\min_{\boldsymbol{x}\in\mathbb{R}^{d_x},\boldsymbol{y}\in\mathbb{R}^{d_y}} \frac{1}{2}\left(f(\boldsymbol{x},\boldsymbol{y}) + \lambda g(\boldsymbol{x},\boldsymbol{y}) - \min_{\boldsymbol{z}\in\mathbb{R}^{d_y}}(-f(\boldsymbol{x},\boldsymbol{z}) + \lambda g(\boldsymbol{x},\boldsymbol{z}))\right). \tag{4}$$

Compared with Eq. (3), this new penalty problem perturbs the lower-level variables $\boldsymbol{y}$ and $\boldsymbol{z}$ in the opposite direction to better cancel out the approximation errors to Problem (1). The connection between bilevel optimization and finite difference approximations was recently established by Chayti & Jaggi (2024) in the context of meta-learning, but their results were limited to symmetric approximations. We extend their findings beyond meta-learning and to general finite difference approximations, addressing their conjecture about broader applicability. In this work, we show that F$^2$SA-2 provably improves the SFO complexity of F$^2$SA from $\tilde{\mathcal{O}}(\epsilon^{-6})$ to $\tilde{\mathcal{O}}(\epsilon^{-5})$ for second-order smooth problems. Moreover, our idea is generalizable for any $p$th-order smooth problems. It is known in numerical analysis there exists the $p$th-order central difference that uses $p$ points to construct an estimator to the derivative of a unitary function with $p$th-order error guarantee, as recalled in Lemma 3.1. Motivated by this fact, we propose the F$^2$SA-$p$ algorithm and show that it enjoys the improved $\tilde{\mathcal{O}}(p\epsilon^{-4-2/p})$ SFO complexity, as formally stated in Theorem 3.1.

Moreover, as formally stated in Theorem 4.1, it is easy to extend the $\Omega(\epsilon^{-4})$ lower bound for SGD (Arjevani et al., 2023) from single-level optimization to bilevel optimization using a fully separable construction that automatically satisfies all our additional smoothness conditions in Definition 2.2. It shows that F$^2$SA-$p$ is optimal up to logarithmic factors when $p = \Omega(\log\epsilon^{-1}/\log\log\epsilon^{-1})$ (see Remark 3.4). We summarize our main results in Table 1 and discuss open problems in the following.

**Open problems.** Our upper bounds improve known results for high-order smooth problems, but our result still has a gap between the $\Omega(\epsilon^{-4})$ lower bound for $p = \mathcal{O}(\log \epsilon^{-1} / \log \log \epsilon^{-1})$. Recently, Kwon et al. (2024a) obtained some preliminary results towards closing this gap for $p = 1$, where they showed an $\Omega(\epsilon^{-6})$ lower bound holds under a more adversarial oracle. But it is still open whether their lower bounds can be extended to standard stochastic oracles. Another open problem is the tightness of the condition number dependency, for which the current upper and lower bounds have a gap of $\Omega(\kappa^9)$ as demonstrated in Table 1. Two recent concurrent works (Ji, 2025; Chen & Zhang, 2025) proposed tighter lower bounds for $p = 1$: Ji (2025) showed a lower bound of $\Omega(\kappa^{5/2}\epsilon^{-4})$ and Chen & Zhang (2025) showed that of $\Omega(\kappa^4\epsilon^{-4})$. However, it is still open to fully close the gap in condition number dependency for both $p = 1$ and $p \geq 2$.

| Method | Smoothness | Reference | Complexity |
|---|---|---|---|
| F²SA | 1st-order | (Kwon et al., 2023) | $\tilde{\mathcal{O}}(\mathrm{poly}(\kappa)\epsilon^{-7})$ |
| F²SA | 1st-order | (Kwon et al., 2024a) | $\tilde{\mathcal{O}}(\mathrm{poly}(\kappa)\epsilon^{-6})$ |
| F²SA | 1st-order | (Chen et al., 2025b) | $\tilde{\mathcal{O}}(\kappa^{12}\epsilon^{-6})$ |
| F²SA-$p$ | 1st-order + $p$th-order in $\boldsymbol{y}$ | Theorem 3.1 | $\tilde{\mathcal{O}}(p\kappa^{9+2/p}\epsilon^{-4-2/p})$ |
| Lower Bound | | (Arjevani et al., 2023) | $\Omega(\epsilon^{-4})$ |

Table 1: The SFO complexity of different methods to find an $\epsilon$-stationary point for $p$th-order smooth first-order bilevel problems with condition number $\kappa$ under standard SGD assumptions.

**Notations.** We use $\|\cdot\|$ to denote the Euclidean norm for vectors and the spectral norm for matrices and tensors. For any set $S$ and functions $g, h : S \to [0, \infty)$ we write $g = \mathcal{O}(h)$ or $h = \Omega(g)$ equivalently if there exists $c > 0$ such that $g(s) \leq ch(s)$ for every $s \in S$. We use $\tilde{\mathcal{O}}(\cdot)$ and $\tilde{\Omega}(\cdot)$ to hide logarithmic factors in $\mathcal{O}(\cdot)$ and $\Omega(\cdot)$. We alsouse $h_1 \lesssim h_2$ to mean $h_1 = \mathcal{O}(h_2)$, $h_1 \gtrsim h_2$ to mean $h_1 = \Omega(h_2)$, and $h_1 \asymp h_2$ to mean that both $h_1 \lesssim h_2$ and $h_1 \gtrsim h_2$ hold. Additional notations for tensors are introduced in Appendix A.

## 2 PRELIMINARIES

The goal of bilevel optimization is to minimize the hyper-objective $\varphi(\boldsymbol{x})$, which is in general non-convex. Since finding a global minimizer of a general nonconvex function requires exponential complexity in the worst case (Nemirovskij & Yudin, 1983, § 1.6), we follow the literature (Carmon et al., 2020; 2021) to consider the task of finding an approximate stationary point.

**Definition 2.1.** *Let $\varphi : \mathbb{R}^{d_x} \to \mathbb{R}$ be the hyper-objective defined in Eq. (1). We say a random variable $\hat{\boldsymbol{x}} \in \mathbb{R}^{d_x}$ is an $\epsilon$-hyper-stationary point if $\mathbb{E}\|\nabla\varphi(\hat{\boldsymbol{x}})\| \leq \epsilon$.*

Next, we introduce the assumptions used in this paper, which ensure the tractability of the above hyper-stationarity. Compared to (Kwon et al., 2023; Chen et al., 2025b), we additionally assume the high-order smoothness in lower-level variable $\boldsymbol{y}$ to achieve further acceleration.

### 2.1 PROBLEM SETUP

First of all, we follow the standard assumptions on SGD (Arjevani et al., 2023) to assume that the stochastic gradient estimators satisfy the following assumption.

**Assumption 2.1.** *There exists stochastic gradient estimators $F(\boldsymbol{x}, \boldsymbol{y})$ and $G(\boldsymbol{x}, \boldsymbol{y})$ such that*

$$\mathbb{E}F(\boldsymbol{x}, \boldsymbol{y}; \xi) = \nabla f(\boldsymbol{x}, \boldsymbol{y}), \quad \mathbb{E}\|F(\boldsymbol{x}, \boldsymbol{y}) - \nabla f(\boldsymbol{x}, \boldsymbol{y})\|^2 \leq \sigma^2;$$
$$\mathbb{E}G(\boldsymbol{x}, \boldsymbol{y}; \zeta) = \nabla g(\boldsymbol{x}, \boldsymbol{y}), \quad \mathbb{E}\|G(\boldsymbol{x}, \boldsymbol{y}) - \nabla g(\boldsymbol{x}, \boldsymbol{y})\|^2 \leq \sigma^2,$$

*where $\sigma > 0$ is the variance of the stochastic gradient estimators. We also partition $F = (F_x, F_y)$ and $G = (G_x, G_y)$ such that $F_x, F_y, G_x, G_y$ are estimators to $\nabla_x f, \nabla_y f, \nabla_x g, \nabla_y g$, respectively.*

Second, we assume that the hyper-objective $\varphi(\boldsymbol{x}) = f(\boldsymbol{x}, \boldsymbol{y}^*(\boldsymbol{x}))$ is lower bounded. Otherwise, any algorithm requires infinite time to find a stationary point. Note that the implicit condition $\inf_{\boldsymbol{x} \in \mathbb{R}^{d_x}} \varphi(\boldsymbol{x}) > -\infty$ below can also be easily implied by a more explicit condition on the lower boundedness of upper-level function $\inf_{\boldsymbol{x} \in \mathbb{R}^{d_x}, \boldsymbol{y} \in \mathbb{R}^{d_y}} f(\boldsymbol{x}, \boldsymbol{y}) > -\infty$.

**Assumption 2.2.** *The hyper-objective defined in Eq. (1) is lower bounded, and we have*

$$\varphi(\boldsymbol{x}_0) - \inf_{\boldsymbol{x} \in \mathbb{R}^{d_x}} \varphi(\boldsymbol{x}) \leq \Delta,$$

*where $\Delta > 0$ is the initial suboptimality gap and we assume $\boldsymbol{x}_0 = \boldsymbol{0}$ without loss of generality.*

Third, we assume the lower-level function $g(\boldsymbol{x}, \boldsymbol{y})$ is strongly convex in $\boldsymbol{y}$. It guarantees the uniqueness of $\boldsymbol{y}^*(\boldsymbol{x})$ and the tractability of the bilevel problem. Although not all the problems in applications satisfy the lower-level strong convexity assumption, it is impossible to derive dimension-free upper bounds when the lower-level problem is only convex (Chen et al., 2024, Theorem 3.2). Hence, we follow most existing works to consider strongly convex lower-level problems.

**Assumption 2.3.** *$g(\boldsymbol{x}, \boldsymbol{y})$ is $\mu$-strongly convex in $\boldsymbol{y}$, i.e., for any $\boldsymbol{y}_1, \boldsymbol{y}_2 \in \mathbb{R}^{d_y}$, we have*

$$g(\boldsymbol{x}, \boldsymbol{y}_2) \geq g(\boldsymbol{x}, \boldsymbol{y}_1) + \langle \nabla_y g(\boldsymbol{x}, \boldsymbol{y}_1), \boldsymbol{y}_2 - \boldsymbol{y}_1 \rangle + \frac{\mu}{2} \|\boldsymbol{y}_1 - \boldsymbol{y}_2\|^2,$$

*where $\mu > 0$ is the strongly convex parameter.*

Fourth, we require the following smoothness assumptions following (Ghadimi & Wang, 2018). According to Eq. (2), these conditions are necessary and sufficient to guarantee the Lipschitz continuity of $\nabla \varphi(\boldsymbol{x})$, which further ensure the tractability of an approximate stationary point of the nonconvex hyper-objective $\varphi(\boldsymbol{x})$ (Zhang et al., 2020; Kornowski & Shamir, 2022).

**Assumption 2.4.** *For the upper-lower function $f$ and lower-level function $g$, we assume that*

1. *$f(\boldsymbol{x}, \boldsymbol{y})$ is $L_0$-Lipschitz in $\boldsymbol{y}$.*

2. *$\nabla f(\boldsymbol{x}, \boldsymbol{y})$ and $\nabla g(\boldsymbol{x}, \boldsymbol{y})$ are $L_1$-Lipschitz jointly in $(\boldsymbol{x}, \boldsymbol{y})$.*

3. *$\nabla^2_{xy} g(\boldsymbol{x}, \boldsymbol{y})$ and $\nabla^2_{yy} g(\boldsymbol{x}, \boldsymbol{y})$ are $L_2$-Lipschitz jointly in $(\boldsymbol{x}, \boldsymbol{y})$.*

We refer to the problem class that jointly satisfies all the above Assumption 2.1, 2.2, 2.3 and 2.4 as first-order smooth bilevel problems, for which (Kwon et al., 2024a; Chen et al., 2025b) showed the F$^2$SA method achieves the $\tilde{\mathcal{O}}(\epsilon^{-6})$ upper complexity bound. In this work, we show an improved bound under the following additional higher-order smoothness assumption on lower-level variable $\boldsymbol{y}$.

**Assumption 2.5** (High order smoothness in $\boldsymbol{y}$). *Given $p \in \mathbb{N}_+$, we assume that*

1. *$\frac{\partial^q}{\partial \boldsymbol{y}^q} \nabla f(\boldsymbol{x}, \boldsymbol{y})$ is $L_{q+1}$-Lipschitz for all $q = 1, \cdots, p-1$.*

2. *$\frac{\partial^{q+1}}{\partial \boldsymbol{y}^{q+1}} \nabla g(\boldsymbol{x}, \boldsymbol{y})$ is $L_{q+2}$-Lipschitz in $\boldsymbol{y}$ for all $q = 1, \cdots, p-1$.*

We refer to problems jointly satisfying all the above assumptions as $p$th-order smooth bilevel problems, and also formally define their condition numbers as follows.

**Definition 2.2** ($p$th-order smooth bilevel problems). *Given $p \in \mathbb{N}_+$, $\Delta > 0$, $L_0, L_1, \cdots, L_{p+1} > 0$, and $\mu \leq L_1$, we use $\mathcal{F}^{nc\text{-}sc}(L_0, \cdots, L_{p+1}, \mu, \Delta)$ to denote the set of all bilevel instances satisfying Assumption 2.2, 2.3, 2.4 and 2.5. For this problem class, we define the largest smoothness constant $\bar{L} = \max_{0 \leq j \leq p} L_j$ and condition number $\kappa = \bar{L}/\mu$.*

All our above assumptions align with (Chen et al., 2025b) except for the additional Assumption 2.5. A classic example of a highly smooth function is the softmax function (Garg et al., 2021, Lemma 2(3)). Therefore, many hyper-parameter tuning problems for logistic regression are provably highly smooth, such that our theory can be applied. We give two examples from (Pedregosa, 2016): the first one aims to learn the optimal weights of each sample in a corrupted training set, and the second one aims to learn the optimal regularizer of each parameter.

**Example 2.1** (Data hyper-cleaning). *Let $\boldsymbol{x} \in \mathbb{R}^n$ parameterize the per-sample weight of a training set with $n$ samples via $\sigma(x_i) = \exp(x_i) / \sum_{i=1}^n \exp(x_i)$ and $\boldsymbol{y} \in \mathbb{R}^d$ be the parameters of a linear*

*model. Let $\ell_{\mathrm{val}}$ be the logistic loss of the linear model on the validation set and $\ell_{\mathrm{tr}}^i$ be the logistic loss on the training sample $i$. The problem aims to solve*

$$\min_{\boldsymbol{x} \in \mathbb{R}^n} \ell_{\mathrm{val}}(\boldsymbol{y}), \quad \text{s.t.} \quad \boldsymbol{y} \in \arg\min_{\boldsymbol{y} \in \mathbb{R}^d} \sum_{i=1}^n \sigma(x_i) \ell_{\mathrm{tr}}^i(\boldsymbol{y}).$$

**Example 2.2** (Learn-to-regularize). *Let $\boldsymbol{x} \in \mathbb{R}^d$ parameterize the regularization matrix via $\boldsymbol{W_x} = \mathrm{diag}(\exp(\boldsymbol{x}))$, and $\boldsymbol{y} \in \mathbb{R}^d$ be the parameters of a linear model. Let $\ell_{\mathrm{val}}$ and $\ell_{\mathrm{tr}}$ be the logistic loss of the linear model on the validation set and training set, respectively. The problem aims to solve*

$$\min_{\boldsymbol{x} \in \mathbb{R}^d} \ell_{\mathrm{val}}(\boldsymbol{y}), \quad \text{s.t.} \quad \boldsymbol{y} \in \arg\min_{\boldsymbol{y} \in \mathbb{R}^d} \ell_{\mathrm{tr}}(\boldsymbol{y}) + \boldsymbol{y}^\top \boldsymbol{W_x} \boldsymbol{y}.$$

## 2.2 COMPARISON TO PREVIOUS WORKS

Before we show our improved upper bound, we first give a detailed discussion on other assumptions made in related works.

**Stochastic Hessian assumption.** Ghadimi & Wang (2018); Ji et al. (2021) assumes the access to a stochastic Hessian estimator $\boldsymbol{H}(\boldsymbol{x}, \boldsymbol{y})$ such that

$$\mathbb{E}\boldsymbol{H}(\boldsymbol{x}, \boldsymbol{y}) = \nabla^2 g(\boldsymbol{x}, \boldsymbol{y}), \quad \mathbb{E}\|\boldsymbol{H}(\boldsymbol{x}, \boldsymbol{y}) - \nabla^2 g(\boldsymbol{x}, \boldsymbol{y})\| \le \sigma^2. \tag{5}$$

According to (Arjevani et al., 2020, Observation 1 and 2), such an assumption is stronger than standard SGD assumptions and equivalent to the mean-squared-smoothness assumption (6) on the lower-level gradient estimator $G$ under the mild condition of $\nabla G(\boldsymbol{x}, \boldsymbol{y}) = \boldsymbol{H}(\boldsymbol{x}, \boldsymbol{y})$. Under this assumption, in conjunction with Assumption 2.2, 2.3, and 2.4, Ghadimi & Wang (2018) proposed the BSA method that can find an $\epsilon$ stationary point of $\varphi(\boldsymbol{x})$ with $\tilde{\mathcal{O}}(\epsilon^{-6})$ SFO complexity and $\tilde{\mathcal{O}}(\epsilon^{-4})$ stochastic HVP complexity. Later, Ji et al. (2021) further improved the SFO complexity term to $\tilde{\mathcal{O}}(\epsilon^{-4})$. Compared to them, we consider the setting where the algorithms only have access to stochastic gradient estimators, and make no assumptions on the stochastic Hessians.

**Mean-squared smoothness assumption.** Besides Assumption 2.1, 2.2, 2.3, 2.4 and the stochastic Hessian assumption (5), Khanduri et al. (2021); Yang et al. (2021; 2023b) further assumes that the stochastic estimators to gradients and Hessians are mean-squared smooth:

$$\begin{aligned} \mathbb{E}\|F(\boldsymbol{x}, \boldsymbol{y}) - F(\boldsymbol{x}', \boldsymbol{y}')\|^2 &\le \bar{L}_1^2 \|(\boldsymbol{x}, \boldsymbol{y}) - (\boldsymbol{x}', \boldsymbol{y}')\|^2, \\ \mathbb{E}\|G(\boldsymbol{x}, \boldsymbol{y}) - G(\boldsymbol{x}', \boldsymbol{y}')\|^2 &\le \bar{L}_1^2 \|(\boldsymbol{x}, \boldsymbol{y}) - (\boldsymbol{x}', \boldsymbol{y}')\|^2, \\ \mathbb{E}\|\boldsymbol{H}(\boldsymbol{x}, \boldsymbol{y}) - \boldsymbol{H}(\boldsymbol{x}', \boldsymbol{y}')\|^2 &\le \bar{L}_2^2 \|(\boldsymbol{x}, \boldsymbol{y}) - (\boldsymbol{x}', \boldsymbol{y}')\|^2. \end{aligned} \tag{6}$$

Under this additional assumption, they proposed faster stochastic methods with upper complexity bound of $\tilde{\mathcal{O}}(\epsilon^{-3})$ via variance reduction (Fang et al., 2018; Cutkosky & Orabona, 2019). In this paper, we only consider the setting without mean-squared smoothness assumptions and study a different acceleration mechanism from variance reduction.

**Jointly high-order smoothness assumption.** Huang et al. (2025) introduced a second-order smoothness assumption similar to but stronger than Assumption 2.5 when $p = 2$. Specifically, they assumed the second-order smoothness jointly in $(\boldsymbol{x}, \boldsymbol{y})$ instead of $\boldsymbol{y}$ only:

$$\begin{aligned} \nabla^2 f(\boldsymbol{x}, \boldsymbol{y}) &\text{ is } L_2\text{-Lipschitz jointly in } (\boldsymbol{x}, \boldsymbol{y}); \\ \nabla^3 g(\boldsymbol{x}, \boldsymbol{y}) &\text{ is } L_3\text{-Lipschitz jointly in } (\boldsymbol{x}, \boldsymbol{y}). \end{aligned} \tag{7}$$

The jointly second-order smoothness (7) ensures that the hyper-objective $\varphi(\boldsymbol{x})$ has Lipschitz continuous Hessians, which further allows the application of known techniques in minimizing second-order smooth objectives. Huang et al. (2025) applied the technique from (Jin et al., 2017; 2021; Xu et al., 2018; Allen-Zhu & Li, 2018) to show that an HVP-based method can find a second-order stationary point in $\tilde{\mathcal{O}}(\epsilon^{-2})$ complexity under the deterministic setting, and in $\tilde{\mathcal{O}}(\epsilon^{-4})$ under the stochastic Hessian assumption (5). Yang et al. (2023a) applied the technique from (Li & Lin, 2023) to accelerate the complexity HVP-based method to $\tilde{\mathcal{O}}(\epsilon^{-1.75})$ in the deterministic setting. Chen et al. (2025b) also proposed a fully first-order method to achieve the same $\tilde{\mathcal{O}}(\epsilon^{-1.75})$ complexity. Compared to these works, our work demonstrates a unique acceleration mechanism in stochastic bilevel optimization that only comes from the high-order smoothness in $\boldsymbol{y}$.

## 3 THE F²SA-$p$ METHOD

To introduce our method, we first recall the prior F²SA method (Kwon et al., 2023) and establish their relationship between finite difference schemes, which further motivates us to design better algorithms by using better finite difference formulas.

### 3.1 HYPER-GRADIENT APPROXIMATION VIA FINITE DIFFERENCE

The core idea of F²SA (Kwon et al., 2023) is to solve the reformulated penalty problem (3) and use the gradient of the penalty function to approximate the true hyper-gradient. To make connections of F²SA to the finite difference method, let us introduce the extra notation $g_\nu$ as the perturbed lower-lever problem with $\boldsymbol{y}_\nu^*(\boldsymbol{x})$ and $\ell_\nu(\boldsymbol{x})$ being its optimal solution and optimal value, respectively:

$$g_\nu(\boldsymbol{x}, \boldsymbol{y}) := \nu f(\boldsymbol{x}, \boldsymbol{y}) + g(\boldsymbol{x}, \boldsymbol{y}),$$
$$\boldsymbol{y}_\nu^*(\boldsymbol{x}) := \arg \min_{\boldsymbol{y} \in \mathbb{R}^{d_y}} g_\nu(\boldsymbol{x}, \boldsymbol{y}),$$
$$\ell_\nu(\boldsymbol{x}) := \min_{\boldsymbol{y} \in \mathbb{R}^{d_y}} g_\nu(\boldsymbol{x}, \boldsymbol{y}),$$

Then we have $\frac{\partial}{\partial \nu}\ell_\nu(\boldsymbol{x})|_{\nu=0} = \lim_{\nu \to 0} \frac{\ell_\nu(\boldsymbol{x})-\ell_0(\boldsymbol{x}))}{\nu} = \lim_{\nu \to 0} f(\boldsymbol{x}, \boldsymbol{y}_\nu^*(\boldsymbol{x})) + \frac{g(\boldsymbol{x}, \boldsymbol{y}_\nu^*(\boldsymbol{x}))-g(\boldsymbol{x}, \boldsymbol{y}^*(\boldsymbol{x}))}{\nu}$. In our notation, we can rewrite (Chen et al., 2025b, Lemma B.3) as $\frac{\partial}{\partial \nu}\ell_\nu(\boldsymbol{x})|_{\nu=0} = \varphi(\boldsymbol{x})$. Similarly, we can also rewrite (Kwon et al., 2023, Lemma 3.1) as

$$\frac{\partial^2}{\partial \nu \partial \boldsymbol{x}}\ell_\nu(\boldsymbol{x})|_{\nu=0} = \frac{\partial^2}{\partial \boldsymbol{x} \partial \nu}\ell_\nu(\boldsymbol{x})|_{\nu=0} = \nabla\varphi(\boldsymbol{x}). \tag{8}$$

Let $\nu = 1/\lambda$ in Eq. (3). Then the fully first-order hyper-gradient estimator (Kwon et al., 2023; Chen et al., 2025b) is exactly using forward difference to approximate $\nabla\varphi(\boldsymbol{x})$, that is,

$$\frac{\frac{\partial}{\partial \boldsymbol{x}}\ell_\nu(\boldsymbol{x}) - \frac{\partial}{\partial \boldsymbol{x}}\ell_0(\boldsymbol{x})}{\nu} \approx \frac{\partial^2}{\partial \nu \partial \boldsymbol{x}}\ell_\nu(\boldsymbol{x})|_{\nu=0} = \nabla\varphi(\boldsymbol{x}). \tag{9}$$

However, the forward difference is not the only way to approximate a derivative. Essentially, it falls into a general class of $p$th-order finite difference (Atkinson & Han, 2005) that can guarantee an $\mathcal{O}(\nu^p)$ approximation error. We restate this known result (with generalization to vector-valued functions) in the following lemma and provide a self-contained proof in Appendix B for completeness.

**Lemma 3.1.** *Assume the function $\psi : \mathbb{R} \to \mathbb{R}^d$ has $C$-Lipschitz continuous $p$th-order derivative. There exist coefficients $\{\alpha_j\}$ such that*

$$\left\| \frac{1}{\nu}\sum_j \alpha_j \psi(j\nu) - \psi'(0) \right\| = \mathcal{O}(C\nu^p).$$

*If $p$ is even, the indices run $j = -p/2, \cdots, p/2$. If $p$ is odd, they run $j = -(p-1)/2, \cdots, (p+1)/2$. Furthermore, all the coefficients satisfy $|j\alpha_j| \leq 1$ for all $j \neq 0$ and $|\alpha_0| \leq \mathbb{I}[p \text{ is odd}]$.*

The explicit formulas for $\alpha_j$ can be found in Appendix B. When $p = 1$, we have $\alpha_0 = -1$, $\alpha_1 = 1$, and we obtain the forward difference estimator $\psi(\nu) - \psi(0)/\nu$; When $p = 2$ we have $\alpha_{-1} = -1/2, \alpha_1 = 1/2$ and we obtain the central difference estimator $(\psi(\nu) - \psi(-\nu))/(2\nu)$. Lemma 3.1 tells us that in general we can always construct a finite difference estimator $\mathcal{O}(\nu^p)$ error with $p$ points for even $p$ or $p + 1$ points for odd $p$ under the given smoothness conditions. Inspired by Lemma 3.1 and Eq. (8) that $\frac{\partial^2}{\partial \nu \partial \boldsymbol{x}}\ell_\nu(\boldsymbol{x})|_{\nu=0} = \nabla\varphi(\boldsymbol{x})$, we propose a fully first-order estimator via a linear combination of $\frac{\partial}{\partial \boldsymbol{x}}\ell_{j\nu}(\boldsymbol{x})$ to achieve $\mathcal{O}(\nu^p)$ approximation error to $\nabla\varphi(\boldsymbol{x})$ given that $\frac{\partial^{p+1}}{\partial \nu^p \partial \boldsymbol{x}}\ell_\nu(\boldsymbol{x})$ is Lipschitz continuous in $\nu$. It further leads to Algorithm 1 that will be formally introduced in the next subsection.

### 3.2 THE PROPOSED ALGORITHM

Due to space limitations, we only present Algorithm 1 designed for even $p$ in the main text. The algorithm for odd $p$ can be designed similarly, and we defer the concrete algorithm to Appendix D.

---

**Algorithm 1** $\text{F}^2\text{SA-}p\ (\boldsymbol{x}_0, \boldsymbol{y}_0)$, even $p$

---

1: $\boldsymbol{y}_0^j = \boldsymbol{y}_0, \ \forall j \in \mathbb{N}$
2: **for** $t = 0, 1, \cdots, T - 1$
3:    **parallel for** $j = -p/2, -p/2 + 1, \cdots, p/2$
4:       $\boldsymbol{y}_t^{j,0} = \boldsymbol{y}_t^j$
5:       **for** $k = 0, 1, \cdots, K - 1$
6:          Sample random i.i.d indexes $\{(\xi_j^y, \zeta_j^y)\}$.
7:          $\boldsymbol{y}_t^{j,k+1} = \boldsymbol{y}_t^{j,k} - \eta_y \left( j\nu F_y(\boldsymbol{x}_t, \boldsymbol{y}_t^{j,k}; \xi_j^y) + G_y(\boldsymbol{x}_t, \boldsymbol{y}_t^{j,k}; \zeta_j^y) \right)$
8:       **end for**
9:       $\boldsymbol{y}_{t+1}^j = \boldsymbol{y}_t^{j,K}$
10:   **end parallel for**
11:   Sample random i.i.d indexes $\{(\xi_i^x, \zeta_i^x)\}_{i=1}^S$.
12:   Let $\{\alpha_j\}_{j=-p/2}^{p/2}$ be the $p$th-order finite difference coefficients defined in Lemma 3.1.
13:   $\Phi_t = \frac{1}{S} \sum_{i=1}^S \sum_{j=-p/2}^{p/2} \alpha_j \left( j F_x(\boldsymbol{x}_t, \boldsymbol{y}_{t+1}^j; \xi_i^x) + \dfrac{G_x(\boldsymbol{x}_t, \boldsymbol{y}_{t+1}^j; \zeta_i^x)}{\nu} \right)$
14:   $\boldsymbol{x}_{t+1} = \boldsymbol{x}_t - \eta_x \Phi_t / \|\Phi_t\|$
15: **end for**

---

Algorithm 1 follows the double-loop structure of $\text{F}^2\text{SA}$ (Chen et al., 2025b; Kwon et al., 2024a) and changes the hyper-gradient estimator to the one introduced in the previous section. Now, we give a more detailed introduction to the procedures of the two loops of $\text{F}^2\text{SA-}p$.

1. In the outer loop, the algorithm first samples a mini-batch with size $S$ and uses Lemma 3.1 to construct $\Phi_t$ via the linear combination of $\frac{\partial}{\partial \boldsymbol{x}} \ell_{j\nu}(\boldsymbol{x}_t)$ for $j = -p/2, \cdots, p/2$ every iteration. After obtaining $\Phi_t$ as an approximation to $\nabla \varphi(\boldsymbol{x}_t)$, the algorithm then performs a normalized gradient descent step $\boldsymbol{x}_{t+1} = \boldsymbol{x}_t - \eta_x \Phi_t / \|\Phi_t\|$ with total $T$ iterations.

2. In the inner loop, the algorithm returns an approximation to $\frac{\partial}{\partial \boldsymbol{x}} \ell_{j\nu}(\boldsymbol{x}_t)$ for all $j = -p/2, \cdots, p/2$. Note that Danskin's theorem indicates $\frac{\partial}{\partial \boldsymbol{x}} \ell_{j\nu}(\boldsymbol{x}_t) = \frac{\partial}{\partial \boldsymbol{x}} g_{j\nu}(\boldsymbol{x}_t, \boldsymbol{y}_{j\nu}^*(\boldsymbol{x}_t))$. It suffices to approximate $\boldsymbol{y}_{j\nu}^*(\boldsymbol{x}_t)$ to sufficient accuracy, which is achieved by taking a $K$-step single-batch SGD subroutine with stepsize $\eta_y$ on each function $g_{j\nu}(\boldsymbol{x}, \cdot)$.

**Remark 3.1** (Effect of normalized gradient step). *Compared to (Chen et al., 2025b; Kwon et al., 2023), the only modification we make to the outer loop is to change the gradient step to a normalized gradient step. The normalization can control the change of $\boldsymbol{y}_{j\nu}^*(\boldsymbol{x}_t)$ and make the analysis of inner loops easier. We believe that all our theoretical guarantees also hold for the standard gradient step via a more involved analysis.*

### 3.3 COMPLEXITY ANALYSIS

This section contains the complexity analysis of Algorithm 1. We first derive the following lemma from the high-dimensional Faà di Bruno formula (Licht, 2024).

**Lemma 3.2.** *Let $\nu \in (0, 1/(2\kappa)]$. For any instance in the $p$th-order smooth bilevel problem class $\mathcal{F}^{nc\text{-}sc}(L_0, \cdots, L_{p+1}, \mu, \Delta)$ as Definition 2.2, $\frac{\partial^{p+1}}{\partial \nu^p \partial \boldsymbol{x}} \ell_\nu(\boldsymbol{x})$ is $\mathcal{O}(\kappa^{2p+1}\bar{L})$-Lipschitz continuous in $\nu$.*

Our result generalizes the prior result for $p = 1$ (Kwon et al., 2023) to any $p \in \mathbb{N}_+$ and also tightens the prior bounds for $p = 2$ (Chen et al., 2025b) as we remark in the following.

**Remark 3.2** (Tighter bounds for $p = 2$). *Note that the variables $\boldsymbol{x}$ and $\nu$ play equal roles in our analysis. Therefore, our result in $p = 2$ essentially implies that $\frac{\partial^3}{\partial \nu \partial \boldsymbol{x}^2} \ell_\nu(\boldsymbol{x})$ is $\mathcal{O}(\kappa^5 \bar{L})$-Lipschitz continuous in $\nu$ around zero, which tightens the $\mathcal{O}(\kappa^6 \bar{L})$ bound of Hessian convergence in (Chen et al., 2025b, Lemma 5.1a) and is of independent interest. The main insight is to avoid the direct calculation of $\nabla^2 \varphi(\boldsymbol{x}) = \frac{\partial^3}{\partial \nu \partial \boldsymbol{x}^2} \ell_\nu(\boldsymbol{x})|_{\nu=0}$ which involves third-order derivatives and makes the analysis more complex, but instead always to analyze it through the limiting point $\lim_{\nu \to 0} \frac{\partial^3}{\partial \nu \partial \boldsymbol{x}^2} \ell_\nu(\boldsymbol{x})$.*

Recall Eq. (8) that $\frac{\partial^2}{\partial\nu\partial\boldsymbol{x}}\ell_\nu(\boldsymbol{x})|_{\nu=0} = \nabla\varphi(\boldsymbol{x})$. Then Lemma 3.2, in conjunction with Lemma 3.1, indicates that the $p$th-order finite difference used in F$^2$SA-$p$ guarantees an $\mathcal{O}(\nu^p)$-approximation error to $\nabla\varphi(\boldsymbol{x})$, which always improves the $\mathcal{O}(\nu)$-error guarantee of F$^2$SA (Kwon et al., 2023; Chen et al., 2025b) for any $p \geq 2$. This improved error guarantee means that we can set $\nu = \mathcal{O}(\epsilon^{1/p})$ to obtain an $\mathcal{O}(\epsilon)$-accurate hyper-gradient estimator to $\nabla\varphi(\boldsymbol{x})$, which further leads to the following improved complexity of our algorithm.

**Theorem 3.1** (Main theorem). *For any instance in the $p$th-order smooth bilevel problem class $\mathcal{F}^{nc\text{-}sc}(L_0, \cdots, L_{p+1}, \mu, \Delta)$ as per Definition 2.2, set the hyper-parameters as*

$$\nu \asymp \min\left\{\frac{R}{\kappa}, \left(\frac{\epsilon}{\bar{L}\kappa^{2p+1}}\right)^{1/p}\right\}, \ \ \eta_x \asymp \frac{\epsilon}{L_1\kappa^3}, \ \ \eta_y \asymp \frac{\nu^2\epsilon^2}{L_1\kappa\sigma^2},$$
$$S \asymp \frac{\sigma^2}{\nu^2\epsilon^2}, \ \ K \asymp \frac{\kappa^2\sigma^2}{\nu^2\epsilon^2}\log\left(\frac{RL_1\kappa}{\nu\epsilon}\right), \ \ T \asymp \frac{\Delta}{\eta_x\epsilon}, \quad (10)$$

*where $R = \|\boldsymbol{y}_0 - \boldsymbol{y}^*(\boldsymbol{x}_0)\|$. Run Algorithm 1 if $p$ is even or Algorithm 2 (in Appendix D) if $p$ is odd. Then we can provably find an $\epsilon$-stationary point of $\varphi(\boldsymbol{x})$ with the total SFO calls upper bounded by*

$$pT(S+K) = \mathcal{O}\left(\frac{p\Delta L_1\bar{L}^{2/p}\sigma^2\kappa^{9+2/p}}{\epsilon^{4+2/p}}\log\left(\frac{RL_1\bar{L}\kappa}{\epsilon}\right)\right).$$

The above theorem shows that the F$^2$SA-$p$ method can achieve the $\tilde{\mathcal{O}}(p\kappa^{9+2/p}\epsilon^{-4-2/p}\log(\kappa/\epsilon))$ SFO complexity for $p$th-order smooth bilevel problems. In the following, we give several remarks on the complexity in different regions of $p$.

**Remark 3.3** (First-order smooth region). *For $p = 1$, our upper bound becomes $\tilde{\mathcal{O}}(\kappa^{11}\epsilon^{-6})$, which improves the $\tilde{\mathcal{O}}(\kappa^{12}\epsilon^{-6})$ bound in (Chen et al., 2025b) by a factor of $\kappa$. The improvement comes from a tighter analysis in the lower-level SGD update and a careful parameter setting.*

**Remark 3.4** (Highly-smooth region). *For $p = \Omega(\log(\kappa/\epsilon)/\log\log(\kappa/\epsilon))$ in Definition 2.2, we can run F$^2$SA-$q$ with $q \asymp \log(\kappa/\epsilon)/\log\log(\kappa/\epsilon)$ and the $\mathcal{O}(q\kappa^9\epsilon^{-4}(\kappa/\epsilon)^{2/q}\log(\kappa/\epsilon))$ complexity in Theorem 3.1 simplifies to $\mathcal{O}(\kappa^9\epsilon^{-4}\log^3(\kappa/\epsilon)/\log\log(\kappa/\epsilon)) = \tilde{\mathcal{O}}(\kappa^9\epsilon^{-4})$, which matches the best-known complexity for HVP-based methods (Ji et al., 2021) under stochastic Hessian assumption (5).*

In the upcoming section, we will derive an $\Omega(\epsilon^{-4})$ lower bound to prove that the F$^2$SA-$p$ is near-optimal in the above highly-smooth region if the condition number $\kappa$ is a constant. We leave the study of optimal complexity for non-constant $\kappa$ to future work.

**Comparison of results for odd $p$ and even $p$.** Note that by Lemma 3.1 when $p$ is odd, we need to use $p+1$ points to construct the estimator, which means the algorithm needs to solve $p+1$ lower-level problems in each iteration to achieve an $\mathcal{O}(\nu^p)$ error guarantee. In contrast, when $p$ is even, $p$ points are enough since the $p$th-order central difference estimator satisfies that $\alpha_0 = 0$. It suggests that even when $p$ is odd, the algorithm designed for odd $p$ may still be better. For instance, the F$^2$SA-2 may always be a better choice than F$^2$SA since its benefits *almost come for free*: (1) it still only needs to solve 2 lower-level problems as the F$^2$SA method, which means the per-iteration complexity remains the same. (2) Although the improved complexity of F$^2$SA-2 relies on the second-order smooth condition, without such a condition, its error guarantee in hyper-gradient estimation will only degenerate to a first-order one, which means it is at least as good as F$^2$SA.

## 4   An $\Omega(\epsilon^{-4})$ Lower Bound

In this section, we prove an $\Omega(\epsilon^{-4})$ lower bound for stochastic bilevel optimization via a reduction to single-level optimization. Our lower bound holds for any randomized algorithms $\mathtt{A}$, which can be defined as a sequence of measurable mappings $\{\mathtt{A}_t\}_{t=1}^T$ that is defined recursively by

$$(\boldsymbol{x}_{t+1}, \boldsymbol{y}_{t+1}) = \mathtt{A}_t\left(r, F(\boldsymbol{x}_0, \boldsymbol{y}_0), G(\boldsymbol{x}_0, \boldsymbol{y}_0)), \cdots, F(\boldsymbol{x}_t, \boldsymbol{y}_t), G(\boldsymbol{x}_t, \boldsymbol{y}_t)\right), \ \ t \in \mathbb{N}_+, \quad (11)$$

where $r$ is a random seed drawn at the beginning to produce the queries, and $F, G$ are the stochastic gradient estimators that satisfy Assumption 2.1. Without loss of generality, we assume that $(\boldsymbol{x}_0, \boldsymbol{y}_0) = (\boldsymbol{0}, \boldsymbol{0})$. Otherwise, we can prove the same lower bound by shifting the functions.

**The construction.**  We construct a separable bilevel instance such that the upper-level function $f(\boldsymbol{x}, \boldsymbol{y}) \equiv f_U(\boldsymbol{x})$ and its stochastic gradient align with the hard instance in (Arjevani et al., 2023), while the lower-level function is the simple quadratic $g(\boldsymbol{x}, y) \equiv g(y) = \mu y^2/2$ with deterministic gradients. We defer the concrete construction to Appendix E. For this separable bilevel instance, we can show that for any randomized algorithm defined in Eq. (11) that uses oracles $(F_U, G)$, the progress in $\boldsymbol{x}$ can be simulated by another randomized algorithm that only uses $F_U$, meaning that the single-level lower bound (Arjevani et al., 2023) also holds.

**Theorem 4.1** (Lower bound). *There exist numerical constants $c > 0$ such that for all $\Delta > 0$, $L_1, \cdots, L_{p+1} > 0$ and $\epsilon \le c\sqrt{L_1\Delta}$, there exists a distribution over the function class $\mathcal{F}^{nc\text{-}sc}(L_0, \cdots, L_{p+1}, \mu, \Delta)$ and the stochastic gradient estimators satisfying Assumption 2.1, such that any randomized algorithm $\mathbb{A}$ defined as Eq. (11) can not find an $\epsilon$-stationary point of $\varphi(\boldsymbol{x}) = f(\boldsymbol{x}, \boldsymbol{y}^*(\boldsymbol{x}))$ in less than $\Omega(\Delta L_1 \sigma^2 \epsilon^{-4})$ SFO calls.*

Below, we give a detailed discussion on the constructions in related works.

**Comparison to other bilevel lower bounds.**  Dagréou et al. (2024) proved lower bounds for finite-sum bilevel optimization via a similar reduction to single-level optimization. However, the direct extension of their construction in the fully stochastic setting gives $f(\boldsymbol{x}, \boldsymbol{y}) = f_U(\boldsymbol{y})$ and $g(\boldsymbol{x}, \boldsymbol{y}) = (\boldsymbol{x} - \boldsymbol{y})^2$, where the high-order derivatives of $f(\boldsymbol{x}, \boldsymbol{y})$ not $\mathcal{O}(1)$-Lipschitz in $\boldsymbol{y}$ and thus violates our assumptions. Kwon et al. (2024a) also proved an $\Omega(\epsilon^{-4})$ lower bound for stochastic bilevel optimization. However, their construction $f(\boldsymbol{x}, y) = y$ and $g(\boldsymbol{x}, y) = (f_U(\boldsymbol{x}) - y)^2$ violate the first-order smoothness of $g(\boldsymbol{x}, y)$ in $\boldsymbol{x}$ when $y$ is far way from $f_U(\boldsymbol{x})$. In this work, we use a fully separable construction to avoid all the aforementioned issues in other works.

## 5 EXPERIMENTS

In this section, we conduct numerical experiments to verify our theory. Following (Grazzi et al., 2020; Ji et al., 2021), we consider the "learn-to-regularize" problem of logistic regression (Example 2.2) on the "20 Newsgroup" dataset, which provably satisfies the highly smooth assumption of any order. The dataset contains 18,000 samples, each sample consists of a feature vector in dimensional $130, 107$ vector and a label that takes a value in $\{1, \cdots, 20\}$. We compare our proposed method $F^2SA$-$p$ with $p \in \{2, 3, 5, 8, 10\}$ with both the previous best fully first-order method $F^2SA$ (Kwon et al., 2023; Chen et al., 2025b) and other Hessian-vector-product-based methods stocBiO (Ji et al., 2021), MRBO and VRBO (Yang et al., 2021). We also include a baseline "w/o Reg" that means the training result of SGD without tuning any regularization. For all the algorithms, we search the other hyperparameters (including $\eta_x, \eta_y, \nu$) in a logarithmic scale with base 10. We run the algorithms with $K = 10$ iterations in the inner loop, and $T = 1000$ iterations in the outer loop, and report the test loss/accuracy *v.s.* the number of outer-loop iterations $t$ in Figure 1. To demonstrate the potential of our methods on nonsmooth nonconvex problems, we also provide additional experiments on a 5-layer multilayer perceptron (MLP) network with ReLU activation in Appendix F. The codes to reproduce our experiments are available online [1].

## 6 CONCLUSIONS AND FUTURE WORKS

This paper proposes a class of fully first-order method $F^2SA$-$p$ that achieves the $\tilde{\mathcal{O}}(p\epsilon^{-4-2/p})$ SFO complexity for $p$th-order smooth bilevel problems. Our result generalized the best-known $\tilde{\mathcal{O}}(\epsilon^{-6})$ result (Kwon et al., 2024a; Chen et al., 2025b) from $p = 1$ to any $p \in \mathbb{N}_+$. We also complement our result with an $\Omega(\epsilon^{-4})$ lower bound to show that our method is near-optimal when $p = \Omega(\log \epsilon^{-1}/\log\log \epsilon^{-1})$. Nevertheless, a gap still exists when $p$ is small, and how to fill it even for the basic setting $p = 1$ is an open problem. Another open problem is whether our theory can be extended our theory to structured nonconvex-nonconvex bilevel problems studied by many recent works (Kwon et al., 2024b; Chen et al., 2024; 2025a; Jiang et al., 2025; Xiao et al., 2023; Xiao & Chen, 2025). In addition, it will also be interesting to further improve the convergence rate of our methods by combining them with variance-reduction (Fang et al., 2018; Cutkosky & Orabona, 2019) or momentum techniques (Fang et al., 2019; Cutkosky & Mehta, 2020).

---

[1]https://github.com/TrueNobility303/F2BA

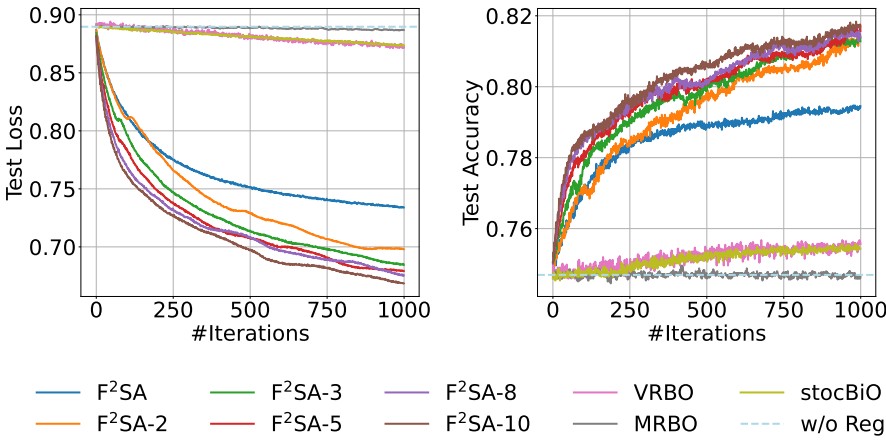

Figure 1: Performances of different algorithms on Example 2.2.

ACKNOWLEDGMENTS

Lesi Chen thanks Jeongyeol Kwon for helpful discussions and Sanyou Mei for pointing out related references. Jingzhao Zhang is supported by Shanghai Qi Zhi Institute Innovation Program, Tsinghua Dushi Funds, and Xiongan AI Institute.

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
