## A  NOTATIONS FOR TENSORS

We follow the notation of tensors used by Kolda & Bader (2009). For two $p$-way tensors $\mathcal{X} \in \mathbb{R}^{n_1 \times n_2 \times \cdots \times n_p}$ and $\mathcal{Y} \in \mathbb{R}^{n_1 \times n_2 \times \cdots \times n_p}$, their inner product $z = \langle \mathcal{X}, \mathcal{Y} \rangle$ is defined as

$$\langle \mathcal{X}, \mathcal{Y} \rangle = \sum_{i_1=1}^{n_1} \sum_{i_2=1}^{n_2} \cdots \sum_{i_q=1}^{n_p} \mathcal{X}_{i_1,i_2,\cdots,i_p} \mathcal{Y}_{i_1,i_2,\cdots,i_p}.$$

For two tensors $\mathcal{X} \in \mathbb{R}^{n_1 \times n_2 \times \cdots n_p}$ and $\mathcal{Y} \in \mathbb{R}^{m_1 \times m_2 \cdots \times m_q}$, their outer product $\mathcal{Z} = \mathcal{X} \otimes \mathcal{Y}$ is a tensor $\mathcal{Z} \in \mathbb{R}^{n_1 \times n_2 \times \cdots \times n_p \times m_1 \times m_2 \times \cdots \times m_q}$ whose elements are defined as

$$(\mathcal{X} \otimes \mathcal{Y})_{i_1,i_2,\cdots,i_p,j_1,j_2,\cdots,j_q} = \mathcal{X}_{i_1,i_2,\cdots,i_p} \mathcal{Y}_{i_1,i_2,\cdots,i_p}.$$

The operator norm of a tensor $\mathcal{X} \in \mathbb{R}^{n_1 \times n_2 \times \cdots \times n_p}$ is defined as

$$\|\mathcal{X}\| = \sup_{\|\boldsymbol{u}_i\|=1, i=1,\cdots,p} \langle \mathcal{X}, \boldsymbol{u}_1 \otimes \boldsymbol{u}_2 \otimes \cdots \otimes \boldsymbol{u}_p \rangle.$$

Equipped with the notion of norm, we say a mapping $\mathcal{T} : \mathbb{R} \to \mathbb{R}^{n_1 \times n_2 \times \cdots \times n_p}$ is $D$-bounded if

$$\|\mathcal{T}(\boldsymbol{x})\| \leq D, \quad \forall \boldsymbol{x} \in \mathbb{R}.$$

We say $\mathcal{T}$ is $C$-Lipschitz continuous if

$$\|\mathcal{T}(\boldsymbol{x}) - \mathcal{T}(\boldsymbol{y})\| \leq C \|\boldsymbol{x} - \boldsymbol{y}\|, \quad \forall \boldsymbol{x}, \boldsymbol{y} \in \mathbb{R}.$$

## B  PROOF OF LEMMA 3.1

*Proof.* If $\psi^{(p)}(\nu)$ is $C$-Lipschitz continuous in $\nu$, then by Taylor's theorem we have

$$\psi(\nu) = \psi(0) + \sum_{k=1}^{p} \frac{(j\nu)^k}{k!} \psi^{(k)}(0) + \mathcal{O}\left(C\nu^{p+1}\right). \tag{12}$$

Now, we analyze the case when $p$ is even or odd separately.

**If $p$ is even.**  The estimator we use is known as the $p$th-order central difference, whose coefficients are known (Khan et al., 2003). Let $n = p/2$. We select coefficients $\{\alpha_j\}_{j=-n}^{n}$ such that

$$\alpha_j = -\alpha_{-j}, \quad \forall j = 0, 1, \cdots, n.$$

Then, summing up Eq. (12) with coefficients $\alpha_j$ gives

$$\frac{1}{\nu} \sum_{j=-n}^{j=n} \alpha_j \psi(j\nu) = \underbrace{2 \sum_{j=1}^{n} \alpha_j \sum_{k=1,3,\cdots}^{n-1} \frac{j^k \nu^{k-1}}{k!} \psi^{(k)}(0)}_{(*)} + \mathcal{O}\left(C\nu^p\right).$$

To let term $(*)$ be equivalent to $\psi'(0)$, we let $\{\alpha_j\}_{j=1}^{n}$ satisfy the following equations:

$$2 \sum_{j=1}^{n} \alpha_j j^k = \mathbf{1}_{k=1}, \quad \forall k = 1, 3, \cdots, n-1,$$

which is equivalent to let $\{j\alpha_j\}_{j=1}^{n}$ satisfy the following linear equation

$$\begin{pmatrix} 1 & 1 & 1 & \cdots & 1 \\ 1^2 & 2^2 & 3^2 & \cdots & n^2 \\ 1^4 & 2^4 & 3^4 & \cdots & n^4 \\ \vdots & \vdots & \vdots & \ddots & \vdots \\ 1^{2(n-1)} & 2^{2(n-1)} & 3^{2(n-1)} & \cdots & n^{2(n-1)} \end{pmatrix} \begin{pmatrix} \alpha_1 \\ 2\alpha_2 \\ 3\alpha_3 \\ \vdots \\ n\alpha_n \end{pmatrix} = \begin{pmatrix} 1/2 \\ 0 \\ 0 \\ \vdots \\ 0 \end{pmatrix}.$$

Now we solve this linear equation to determine the values of $\{\alpha_j\}_{j=1}^n$. Let $\boldsymbol{A}$ be the coefficient matrix of this linear equation, and let $\boldsymbol{A}_j$ be the matrix such that the $j$th column of $\boldsymbol{A}$ is replaced by the standard unit vector $(1, 0, \cdots, 0)^\top$. By Cramer's rule, we have

$$2j\alpha_j = \frac{\det(\boldsymbol{A}_j)}{\det(\boldsymbol{A})}, \quad j = 1, \cdots, n.$$

By observation, we can find that both $\boldsymbol{A}$ and $\boldsymbol{A}_j$ are Vandermonde matrices. Therefore, we can explicitly calculate both $\det(\boldsymbol{A})$ and $\det(\boldsymbol{A}_j)$ according to the determinant formula of Vandermonde matrices, which leads to

$$2j\alpha_j = \frac{(-1)^{j-1} \cdot ((j-1)!)^2 \cdot (n!)^2 \cdot j! \cdot (2j)!}{(j!)^2 \cdot (j-1)! \cdot (n-j)! \cdot (2j-1)! \cdot (n+j)!} = \frac{2(-1)^{j-1}(n!)^2}{(n-j)! \cdot (n+j)!}.$$

Therefore, we have

$$\alpha_j = \frac{(-1)^{j-1}(n!)^2}{j \cdot (n-j)! \cdot (n+j)!},$$

from which it is clear that $|\alpha_j| \le 1/j$.

**If $p$ is odd.** Instead of using the known $p$th-order forward difference (Khan et al., 2003) for which we find that the coefficients will be exponentially large in $p$, we motivate from the $p$th-order central difference above to obtain a stable estimator by leveraging negative points. Let $n = (p+1)/2$. We select coefficients $\{\alpha_j\}_{j=1-n}^n$ that satisfy the constraint $\sum_{j=1-n}^n \alpha_j = 0$. Then, summing up Eq. (12) with coefficients $\alpha_j$ gives

$$\frac{1}{\nu} \sum_{j=1-n}^{j=n} \alpha_j \psi(j\nu) = \underbrace{\sum_{j=1-n}^{j=n} \alpha_j \sum_{k=1}^p \frac{j^k \nu^{k-1}}{k!} \psi^{(k)}(0)}_{(*)} + \mathcal{O}\left(C\nu^p\right).$$

To let term $(*)$ be equivalent to $\psi'(0)$, we let $\{\alpha_j\}_{j=1-n}^n$ satisfy the following equations:

$$\sum_{j=1-n}^n \alpha_j j^k = \mathbf{1}_{k=1}, \quad \forall k = 1, 2, \cdots, p,$$

which is equivalent to let $\{\hat{\alpha}_j\}_{j=(1-n), j \ne 0}^n$ satisfy the following linear equation

$$\begin{pmatrix} 1 & 1 & \cdots & 1 & 1 & \cdots & 1 \\ 1-n & 2-n & \cdots & -1 & 1 & \cdots & n \\ \vdots & \vdots & \vdots & \ddots & \vdots & \vdots & \ddots & \vdots \\ (1-n)^{2n+1} & (2-n)^{2n+1} & \cdots & (-1)^{2n+1} & 1^{2n+1} & \cdots & n^{2n+1} \end{pmatrix} \begin{pmatrix} \hat{\alpha}_{1-n} \\ \hat{\alpha}_{2-n} \\ \vdots \\ \hat{\alpha}_n \end{pmatrix} = \begin{pmatrix} 1 \\ 0 \\ \vdots \\ 0 \end{pmatrix},$$

where we denote $\hat{\alpha}_j = j\alpha_j$ for $j = 1-n, \cdots, -1$ and $1, \cdots, n$. Now we solve this linear equation to determine the values of $\{\hat{\alpha}_j\}$. Let $\boldsymbol{A}$ be the coefficient matrix of this linear equation, and let $\boldsymbol{A}_j$ be the matrix such that the $j$th column of $\boldsymbol{A}$ is replaced by the standard unit vector $(1, 0, \cdots, 0)^\top$. By Cramer's rule, we have

$$\hat{\alpha}_j = \frac{\det(\boldsymbol{A}_j)}{\det(\boldsymbol{A})}, \quad j = 1, \cdots, n.$$

Similar to the case of even $p$, both $\boldsymbol{A}$ and $\boldsymbol{A}_j$ are Vandermonde matrices. Therefore, we can explicitly calculate both $\det(\boldsymbol{A})$ and $\det(\boldsymbol{A}_j)$ according to the determinant formula of Vandermonde matrices. Then, for $j = 1, \cdots, n$, we can obtain that

$$\alpha_j = \frac{\hat{\alpha}_j}{j} = \frac{(-1)^{j-1}(j-1)!(n-1)!n!j!}{j \cdot j!(j-1)!(n+j-1)!(n-j)!} = \frac{(-1)^{j-1}(n-1)!n!}{j(n+j-1)!(n-j)!}.$$

Similarly, for $j = 1, \cdots, n-1$, we can obtain that

$$\alpha_{-j} = \frac{\hat{\alpha}_{-j}}{-j} = \frac{(-1)^j(n-1)!(j-1)!n!j!}{j \cdot j!(n+j-1)!(j-1)!(n+j)!} = \frac{(-1)^j(n-1)!n!}{j(n-j-1)!(n+j)!}.$$

Therefore, it is easy to see that $|\alpha_j| \leq 1/j$ for $j = 1, \cdots, n$, and $|\alpha_{-j}| \leq 1/j$ for $j = 1, \cdots, n-1$. Finally, we calculate $\alpha_0$ from the constraint $\sum_{j=1-n}^{n} \alpha_j = 0$, which leads to

$$\alpha_0 = -\underbrace{\sum_{j=1}^{n} \frac{(-1)^{j-1}(n-1)!n!}{j(n+j-1)!(n-j)!}}_{S_1} - \underbrace{\sum_{j=1}^{n-1} \frac{(-1)^j(n-1)!n!}{j(n-j-1)!(n+j)!}}_{S_2}. \tag{13}$$

We claim that $\alpha_0 = -1/n$ and hence $|\alpha_0| \leq 1$. We prove our claim by calculating the values of $S_1$ and $S_2$ to obtain $\alpha_0$. For $S_1$, we have

$$S_1 = \sum_{j=1}^{n} (-1)^{j-1} \binom{n}{j} \frac{(n-1)!(j-1)!}{(n+j-1)!}.$$

The fraction on the right is the Beta function $B(j, n)$, which can be represented as the integral $B(j, n) = \int_0^1 x^{j-1}(1-x)^{n-1}\mathrm{d}x$. Therefore,

$$S_1 = \sum_{j=1}^{n} (-1)^{j-1} \binom{n}{j} \int_0^1 x^{j-1}(1-x)^{n-1}\mathrm{d}x$$

$$= \int_0^1 (1-x)^{n-1} \left( \sum_{j=1}^{n} (-1)^{j-1} \binom{n}{j} x^{j-1} \right) \mathrm{d}x$$

$$= \int_0^1 \frac{(1-x)^{n-1}}{x} \left( \sum_{j=1}^{n} (-1)^{j-1} \binom{n}{j} x^j \right) \mathrm{d}x.$$

Substituting the binomial expansion $(1-x)^n = 1 + \sum_{j=0}^{n} \binom{n}{j}(-x)^j$, we then have

$$S_1 = \int_0^1 \frac{(1-x)^{n-1}}{x} \left( 1 - (1-x)^n \right) \mathrm{d}x.$$

Let $y = 1 - x$. We then have

$$S_1 = \int_0^1 \frac{y^{n-1}}{1-y}(1-y^n)\mathrm{d}y.$$

Substituting the geometric series sum $\frac{1-y^n}{1-y} = \sum_{k=0}^{n-1} y^k$, we then have

$$S_1 = \int_0^1 y^{n-1} \left( \sum_{k=0}^{n-1} y^k \right) \mathrm{d}y = \int_0^1 \sum_{k=0}^{n-1} y^{n+k-1}\mathrm{d}y = \sum_{k=0}^{n-1} \int_0^1 y^{n+k-1}\mathrm{d}y = \sum_{k=0}^{n-1} \frac{1}{n+k}.$$

Following similar steps, we can also obtain that

$$S_2 = -\sum_{k=0}^{n-2} \frac{1}{n+k+1}.$$

Now, for $\alpha_0 = -S_1 - S_2$, the summation terms cancel out perfectly, which leads to $\alpha_0 = -1/n$.

$\square$

## C  PROOF OF LEMMA 3.2

The proof relies on the high-dimensional version of the Faà di Bruno formula. To formally state the result, we define the following notions. For a mapping $\mathcal{T} : \mathbb{R}^m \to \mathbb{R}^{n_1 \times \cdots \times n_q}$, we define its $k$th-order directional derivative evaluated at $\boldsymbol{z} \in \mathbb{R}^m$ along the direction $(\boldsymbol{u}_1, \cdots, \boldsymbol{u}_k)$ as

$$\nabla_{\boldsymbol{u}_1, \cdots, \boldsymbol{u}_k}^{k} \mathcal{T}_{|\boldsymbol{z}} = \nabla^k \mathcal{T}_{|\boldsymbol{z}}(\boldsymbol{u}_1, \cdots, \boldsymbol{u}_k).$$

We let the symmetric products of $\boldsymbol{u}_1, \cdots, \boldsymbol{u}_k$ as

$$\boldsymbol{u}_1 \vee \boldsymbol{u}_2 \vee \cdots \vee \boldsymbol{u}_k = \frac{1}{k!} \sum_{\pi \in \mathrm{Perm}(k)} \boldsymbol{u}_{\pi(1)} \otimes \boldsymbol{u}_{\pi(2)} \otimes \cdots \otimes \boldsymbol{u}_{\pi(k)},$$

where $\mathrm{Perm(k)}$ denotes the set of permutations of $\{1, 2, \cdots, k\}$. Also, we define the set of all (unordered) partitions of a set $A$ into $k$ pairwise disjoint non-empty sets as

$$\mathcal{P}(A, k) = \left\{ \boldsymbol{P} = (P_1, \cdots, P_k) \subseteq \mathcal{B}(A) \mid A = \cup_{j=1}^k P_j; \; \emptyset \notin \boldsymbol{P}; \; P_i \cap P_j = \emptyset, \forall i < j \right\},$$

where $\mathcal{B}(A)$ is the power set of $A$, *i.e.*, the set of all subsets of $A$. We also abbreviate $\mathcal{P}(\{1 : q\}, k)$ as $\mathcal{P}(q, k)$. Using the above notions, we have the following result.

**Lemma C.1** ((Licht, 2024, Proposition 3.1)). *Let $\mathcal{T}_1$ and $\mathcal{T}_2$ be two mappings. If $\mathcal{T}_1$ and $\mathcal{T}_2$ are $k$-times differentiable at the point $\boldsymbol{z}$ and $\mathcal{T}_1(\boldsymbol{z})$, respectively, then the composite mapping $\mathcal{T}_2 \circ \mathcal{T}_1$ is $k$-times differentiable at the point $\boldsymbol{z}$ and we have*

$$\nabla^q (\mathcal{T}_2 \circ \mathcal{T}_1)_{|\boldsymbol{z}} (\vee_{i=1}^q \boldsymbol{u}_i) = \sum_{\substack{1 \le k \le q, \\ \boldsymbol{P} \in \mathcal{P}(q, k)}} \nabla^k \mathcal{T}_{2|\mathcal{T}_1(\boldsymbol{z})} \left( \nabla^{|P_1|} \mathcal{T}_{1|\boldsymbol{z}} (\vee_{i \in P_1} \boldsymbol{u}_i), \cdots \nabla^{|P_k|} \mathcal{T}_{1|\boldsymbol{z}} (\vee_{i \in P_k} \boldsymbol{u}_i) \right).$$

Recall Danskin's theorem that $\frac{\partial}{\partial \boldsymbol{x}} \ell_\nu(\boldsymbol{x}) = \frac{\partial}{\partial \boldsymbol{x}} g_\nu(\boldsymbol{x}, \boldsymbol{y}_\nu^*(\boldsymbol{x}))$. We can apply Lemma C.1 with $\mathcal{T}_1 = \boldsymbol{y}_\nu^*(\boldsymbol{x})$ and $\mathcal{T}_1 = \frac{\partial}{\partial \boldsymbol{x}} g_\nu(\boldsymbol{x}, \boldsymbol{y})$ to obtain that

$$\frac{\partial^{q+1}}{\partial \nu^q \partial \boldsymbol{x}} \ell_\nu(\boldsymbol{x}) = \sum_{\substack{1 \le k \le q, \\ \boldsymbol{P} \in \mathcal{P}(q, k)}} \frac{\partial^{k+1}}{\partial \boldsymbol{y}^k \partial \boldsymbol{x}} g_\nu(\boldsymbol{x}, \boldsymbol{y}_\nu^*(\boldsymbol{x})) \left( \frac{\partial^{|P_1|}}{\partial \nu^{|P_1|}} \boldsymbol{y}_\nu^*(\boldsymbol{x}), \cdots, \frac{\partial^{|P_k|}}{\partial \nu^{|P_k|}} \boldsymbol{y}_\nu^*(\boldsymbol{x}) \right). \tag{14}$$

Symmetrically, using the first-order optimality condition $\frac{\partial}{\partial \boldsymbol{y}} g_\nu(\boldsymbol{x}, \boldsymbol{y}_\nu^*(\boldsymbol{x})) = 0$ and where the first identity uses the Lemma C.1 with $\mathcal{T}_1 = \boldsymbol{y}_\nu^*(\boldsymbol{x})$ and $\mathcal{T}_1 = \frac{\partial}{\partial \boldsymbol{y}} g_\nu(\boldsymbol{x}, \boldsymbol{y})$ yields that

$$0 = \sum_{\substack{1 \le k \le q, \\ \boldsymbol{P} \in \mathcal{P}(q, k)}} \frac{\partial^{k+1}}{\partial \boldsymbol{y}^{k+1}} g_\nu(\boldsymbol{x}, \boldsymbol{y}_\nu^*(\boldsymbol{x})) \left( \frac{\partial^{|P_1|}}{\partial \nu^{|P_1|}} \boldsymbol{y}_\nu^*(\boldsymbol{x}), \cdots, \frac{\partial^{|P_k|}}{\partial \nu^{|P_k|}} \boldsymbol{y}_\nu^*(\boldsymbol{x}) \right). \tag{15}$$

Since $\mathcal{P}(q, 1)$ contains only one element, the above identity implies that

$$\frac{\partial^q}{\partial \nu^q} \boldsymbol{y}_\nu^*(\boldsymbol{x}) = - \left( \nabla_{yy}^2 g_\nu(\boldsymbol{x}, \boldsymbol{y}_\nu^*(\boldsymbol{x})) \right)^{-1} \sum_{\substack{2 \le k \le q, \\ \boldsymbol{P} \in \mathcal{P}(q, k)}} \boldsymbol{w}_{k, \boldsymbol{P}},$$

$$\text{where } \boldsymbol{w}_{k, \boldsymbol{P}} = \frac{\partial^{k+1}}{\partial \boldsymbol{y}^{k+1}} g_\nu(\boldsymbol{x}, \boldsymbol{y}_\nu^*(\boldsymbol{x})) \left( \frac{\partial^{|P_1|}}{\partial \nu^{|P_1|}} \boldsymbol{y}_\nu^*(\boldsymbol{x}), \cdots, \frac{\partial^{|P_k|}}{\partial \nu^{|P_k|}} \boldsymbol{y}_\nu^*(\boldsymbol{x}) \right). \tag{16}$$

Based on Eq. (16), we can prove by induction that $\frac{\partial^q}{\partial \nu^q} \boldsymbol{y}_\nu^*(\boldsymbol{x})$ is $\mathcal{O}(\kappa^{2q+1})$-Lipschitz continuous in $\nu$ for all $q = 0, \cdots, p$. The induction base for $q = 0, 1$ is already proved by Chen et al. (2025b).

**Lemma C.2** (Chen et al. (2025b, Lemma B.2 and B.5)). *Let $\nu \in (0, 1/(2\kappa)]$. Under Assumption 2.3 and 2.4, $\boldsymbol{y}_\nu^*(\boldsymbol{x})$ and $\frac{\partial}{\partial \nu} \boldsymbol{y}_\nu^*(\boldsymbol{x})$ is $\mathcal{O}(\kappa)$- and $\mathcal{O}(\kappa^3)$-Lipschitz continuous in $\nu$, respectively.*

Since Eq. (16) also involves $(\nabla_{yy}^2 g_\nu(\boldsymbol{x}, \boldsymbol{y}_\nu^*(\boldsymbol{x})))^{-1}$, we also need the following lemma that gives its boundedness and Lipschitz continuity constants.

**Lemma C.3** (Chen et al. (2025b, Lemma B.1 and Eq. 18)). *Let $\nu \in (0, 1/(2\kappa)]$. Under Assumption 2.3 and 2.4, $(\nabla_{yy} g_\nu(\boldsymbol{x}, \boldsymbol{y}_\nu^*(\boldsymbol{x})))^{-1}$ is $2/\mu$-bounded and $\mathcal{O}(\kappa^2/\mu)$-Lipschitz continuous in $\nu$.*

In the remaining proofs, we will use Eq. (16) prove by induction that $\frac{\partial^q}{\partial \nu^q} \boldsymbol{y}_\nu^*(\boldsymbol{x})$ is $\mathcal{O}(\kappa^{2q+1})$-Lipschitz continuous in $\nu$, then we can easily use Eq. (14) to show that $\frac{\partial^{q+1}}{\partial \nu^q \partial \boldsymbol{x}} \ell_\nu(\boldsymbol{x})$ is $\mathcal{O}(\kappa^{2q+1} \bar{L})$-Lipschitz continuous in $\nu$ for all $q = 0, \cdots, p$. Note that the computational graph of either $\frac{\partial^q}{\partial \nu^q} \boldsymbol{y}_\nu^*(\boldsymbol{x}))$ or $\frac{\partial^{q+1}}{\partial \nu^q \partial \boldsymbol{x}} \ell_\nu(\boldsymbol{x})$ in Eq. (14) or (16) defines a tree, where the root is output, the leaves are inputs, and the other nodes are the intermediate results in the computation. We can analyze the Lipschitz continuities of all the nodes from bottom to top using the following lemma.

**Lemma C.4** (Luo et al. (2022, Lemma 12)). *Let $\mathcal{T}_1$ and $\mathcal{T}_2$ be two tensor-to-tensor mappings. If $\mathcal{T}_1$ is $D_1$-bounded and $C_1$-Lipschitz continuous, $\mathcal{T}_2$ is $D_2$-bounded and $C_2$-Lipschitz continuous, then the product mapping $\mathcal{T}_1 \times \mathcal{T}_2$ is $D_1 D_2$-bounded and $(C_1 D_2 + C_2 D_1)$-Lipschitz continuous.*

*Proof of Lemma 3.2.* Now, we formally begin to prove by induction that $\frac{\partial^q}{\partial \nu^q} \boldsymbol{y}^*_\nu(\boldsymbol{x})$ is $\mathcal{O}(\kappa^{2q+1})$-Lipschitz continuous in $\nu$ for all $q = 0, \cdots, p$. Recall that the induction base follows Lemma C.2. In the following, we use the induction hypothesis that $\frac{\partial^k}{\partial \nu^k} \boldsymbol{y}^*_\nu(\boldsymbol{x})$ is $\mathcal{O}(\kappa^{2k+1})$-Lipschitz continuous in $\nu$ for all $k = 0, \cdots, q-1$ to prove that $\frac{\partial^q}{\partial \nu^q} \boldsymbol{y}^*_\nu(\boldsymbol{x})$ is $\mathcal{O}(\kappa^{2q+1})$-Lipschitz continuous in $\nu$. We know that $\frac{\partial^{k+1}}{\partial \boldsymbol{y}^{k+1}} g_\nu(\boldsymbol{x}, \boldsymbol{y}^*_\nu(\boldsymbol{x}))$ is $\mathcal{O}(\bar{L})$-bounded and $\mathcal{O}(\kappa \bar{L})$-Lipschitz continuous in $\nu$. Therefore, we can use Lemma C.4 to conclude that each $\boldsymbol{w}_{k,\boldsymbol{P}}$ is $\mathcal{O}(\kappa^{\sum_{j=1}^k (2|P_j|-1)} \bar{L}) = \mathcal{O}(\kappa^{2q-k} \bar{L})$-bounded and $\mathcal{O}(\bar{L} \cdot \kappa^{2q-k+2} + \kappa \bar{L} \cdot \kappa^{2q-k}) = \mathcal{O}(\kappa^{2q-k+2} \bar{L})$-Lipschitz continuous in $\nu$. It further implies that the summation $\boldsymbol{w} := \sum_{2 \le k \le q, \boldsymbol{P} \in \mathcal{P}(q,k)} \boldsymbol{w}_{k,\boldsymbol{P}}$ is $\mathcal{O}(\kappa^{2q-2} \bar{L})$-bounded and $\mathcal{O}(\kappa^{2q} \bar{L})$-Lipschitz continuous in $\nu$. Then, we can recall Lemma C.3 that $(\nabla_{yy} g_\nu(\boldsymbol{x}, \boldsymbol{y}^*_\nu(\boldsymbol{x})))^{-1}$ is $2/\mu$-bounded and $\mathcal{O}(\kappa^2/\mu)$-Lipschitz continuous in $\nu$, and use Eq. (16) to finish the induction that $\frac{\partial^q}{\partial \nu^q} \boldsymbol{y}^*_\nu(\boldsymbol{x}) = -\left(\nabla^2_{yy} g_\nu(\boldsymbol{x}, \boldsymbol{y}^*_\nu(\boldsymbol{x}))\right)^{-1} \boldsymbol{w}$ is $\mathcal{O}(\kappa^{2q+1})$-Lipschitz continuous in $\nu$ for all $q = 0, \cdots, p$. Finally, by analogy with the similarity of Eq. (14) and (16), we can follow the same analysis to show that $\frac{\partial^{q+1}}{\partial \nu^q \partial \boldsymbol{x}} \ell_\nu(\boldsymbol{x})$ is $\mathcal{O}(\kappa^{2q+1} \bar{L})$-Lipschitz continuous in $\nu$ for all $= 0, \cdots, p$. $\square$

# D PROOF OF THEOREM 3.1

---

**Algorithm 2** F$^2$SA-$p$ $(\boldsymbol{x}_0, \boldsymbol{y}_0)$, odd $p$

---

1: $\boldsymbol{y}_0^j = \boldsymbol{y}_0, \ \forall j \in \mathbb{N}$
2: **for** $t = 0, 1, \cdots, T-1$
3:    **parallel for** $j = -(p-1)/2, \cdots, (p+1)/2$
4:       $\boldsymbol{y}_t^{j,0} = \boldsymbol{y}_t^j$
5:       **for** $k = 0, 1, \cdots, K-1$
6:          Sample random i.i.d indexes $\{(\xi_j^y, \zeta_j^y)\}$.
7:          $\boldsymbol{y}_t^{j,k+1} = \boldsymbol{y}_t^{j,k} - \eta_y \left( j\nu F_y(\boldsymbol{x}_t, \boldsymbol{y}_t^{j,k}; \xi_j^y) + G_y(\boldsymbol{x}_t, \boldsymbol{y}_t^{j,k}; \zeta_j^y) \right)$
8:       **end for**
9:       $\boldsymbol{y}_{t+1}^j = \boldsymbol{y}_t^{j,K}$
10:    **end parallel for**
11:    Sample random i.i.d indexes $\{(\xi_i^x, \zeta_i^x)\}_{i=1}^S$.
12:    Let $\{\alpha_j\}_{j=-(p-1)/2}^{(p+1)/2}$ be the $p$th-order finite difference coefficients defined in Lemma 3.1.
13:    $\Phi_t = \frac{1}{S} \sum_{i=1}^S \sum_{j=-(p-1)/2}^{(p+1)/2} \alpha_j \left( j F_x(\boldsymbol{x}_t, \boldsymbol{y}_{t+1}^j; \xi_i^x) + \frac{G_x(\boldsymbol{x}_t, \boldsymbol{y}_{t+1}^j; \zeta_i^x)}{\nu} \right)$
14:    $\boldsymbol{x}_{t+1} = \boldsymbol{x}_t - \eta_x \Phi_t / \|\Phi_t\|$
15: **end for**

---

In the main text, we only present the algorithm when $p$ is even. The algorithm when $p$ is odd follows a similar design, which is presented in Algorithm 2 for completeness. Our algorithms consist of a double loop, where the outer loop performs normalized SGD (NSGD) and the inner loop performs SGD. Before we give the formal proof, we first recall the convergence result for (N)SGD.

**Lemma D.1** (Cutkosky & Mehta (2020, Lemma 2)). *Consider the NSGD update $\boldsymbol{x}_{t+1} = \boldsymbol{x}_t - \eta F_t / \|F_t\|$ to optimize a function $f : \mathbb{R}^d \to \mathbb{R}$ with $L$-Lipschitz continuous gradients. We have*

$$\frac{1}{T} \sum_{t=0}^{T-1} \mathbb{E}\|\nabla f(\boldsymbol{x}_t))\| \le \frac{3(f(\boldsymbol{x}_0) - \inf_{\boldsymbol{x} \in \mathbb{R}^d} f(\boldsymbol{x}))}{\eta T} + \frac{3L\eta}{2} + \frac{8}{T} \sum_{t=0}^{T-1} \mathbb{E}\|F_t - \nabla f(\boldsymbol{x}_t)\|.$$

**Lemma D.2** (Kwon et al. (2024a, Lemma C.1)). *Consider the SGD update $\boldsymbol{x}_{t+1} = \boldsymbol{x}_t - \eta F_t$ to optimize a $\mu$-strongly convex function $f : \mathbb{R}^d \to \mathbb{R}$ with $L$-Lipschitz continuous gradients. Let $\boldsymbol{x}^* = \arg\min_{\boldsymbol{x}\in\mathbb{R}^d} f(\boldsymbol{x})$ be the unique minimizer to $f$. Suppose $F_t$ is an unbiased estimator to $\nabla f(\boldsymbol{x}_t)$ with variance bounded by $\sigma^2$. Setting $\eta < 2/(\mu + L)$, we have*

$$\mathbb{E}\|\boldsymbol{x}_t - \boldsymbol{x}^*\|^2 \leq (1 - \mu\eta)^t \|\boldsymbol{x}_0 - \boldsymbol{x}^*\|^2 + \frac{\eta\sigma^2}{\mu}.$$

The following two lemmas are also useful in the analysis.

**Lemma D.3** (Chen et al. (2025b, Lemma 4.1)). *Under Assumption 2.3, and 2.4, the hyper-objective $\varphi(\boldsymbol{x}) = f(\boldsymbol{x}, \boldsymbol{y}^*(\boldsymbol{x}))$ is differentiable and has $L_\varphi = \mathcal{O}(\bar{L}\kappa^3)$-Lipschitz continuous gradients.*

**Lemma D.4** (Chen et al. (2025b, Lemma B.6)). *Let $\nu \in (-1/\kappa, 1/\kappa)$. Under Assumption 2.3, and 2.4, the optimal (perturbed) lower-level solution mapping $\boldsymbol{y}_\nu^*(\boldsymbol{x}) = \arg\min_{\boldsymbol{y}\in\mathbb{R}^{d_y}} \ell_v(\boldsymbol{x}, \boldsymbol{y})$ is $4\kappa$-Lipschitz continuous in $\boldsymbol{x}$.*

Now we are ready to prove Theorem 3.1.

*Proof of Theorem 3.1.* We separately consider the complexity for the outer loop and the inner loop.

**Outer Loop.** According to Lemma D.3, the hyper-objective $\varphi(\boldsymbol{x})$ has $L_\varphi = \mathcal{O}(\bar{L}\kappa^3)$-Lipschitz continuous gradients. If we can guarantee the condition

$$\mathbb{E}\|\Phi_t - \nabla\varphi(\boldsymbol{x}_t)\| \leq \frac{\epsilon}{32}, \quad t = 0, \cdots, T-1, \tag{17}$$

then we can further set $\eta_x = \epsilon/6L_\varphi$ and apply Lemma D.1 to conclude that the algorithm can provably find an $\epsilon$-stationary point of $\varphi(\boldsymbol{x})$ in $T = \lceil 6\Delta/\epsilon\eta_x \rceil = \mathcal{O}(\Delta L_1 \kappa^3 \epsilon^{-2})$ outer iterations.

**Inner Loop.** From the above analysis, the remaining goal is to show that the inner loop always returns $\Phi_t$ satisfying Eq. (17), which requires $\mathbb{E}\|\Phi_t - \nabla\varphi(\boldsymbol{x}_t)\| = \mathcal{O}(\epsilon)$ for all $t = 0, \cdots, T-1$. Note that the setting of mini-batch size $S = \Omega\left(\sigma^2/\nu^2\epsilon^2\right)$ ensures that

$$\begin{cases} \mathbb{E}\left\|\Phi_t - \sum_{j=-p/2}^{p/2} \alpha_j \left( j\nabla_x f(\boldsymbol{x}_t, \boldsymbol{y}_{t+1}^j) + \frac{\nabla_x g(\boldsymbol{x}_t, \boldsymbol{y}_{t+1}^j)}{\nu} \right)\right\| = \mathcal{O}(\epsilon), & p \text{ is even}; \\[3mm] \mathbb{E}\left\|\Phi_t - \sum_{j=-(p-1)/2}^{(p+1)/2} \alpha_j \left( j\nabla_x f(\boldsymbol{x}_t, \boldsymbol{y}_{t+1}^j) + \frac{\nabla_x g(\boldsymbol{x}_t, \boldsymbol{y}_{t+1}^j)}{\nu} \right)\right\| = \mathcal{O}(\epsilon), & p \text{ is odd}. \end{cases}$$

By Lemma 3.2 and Lemma 3.1, setting $\nu = \mathcal{O}((\epsilon/\bar{L}\kappa^{2p+1})^{1/p})$ can ensure that

$$\begin{cases} \left\|\nabla\varphi(\boldsymbol{x}_t) - \sum_{j=-p/2}^{p/2} \alpha_j \left( j\nabla_x f(\boldsymbol{x}_t, \boldsymbol{y}_{j\nu}^*(\boldsymbol{x}_t)) + \frac{\nabla_x g(\boldsymbol{x}_t, \boldsymbol{y}_{j\nu}^*(\boldsymbol{x}_t))}{\nu} \right)\right\| = \mathcal{O}(\epsilon), & p \text{ is even}; \\[3mm] \left\|\nabla\varphi(\boldsymbol{x}_t) - \sum_{j=-(p-1)/2}^{(p+1)/2} \alpha_j \left( j\nabla_x f(\boldsymbol{x}_t, \boldsymbol{y}_{j\nu}^*(\boldsymbol{x}_t)) + \frac{\nabla_x g(\boldsymbol{x}_t, \boldsymbol{y}_{j\nu}^*(\boldsymbol{x}_t))}{\nu} \right)\right\| = \mathcal{O}(\epsilon), & p \text{ is odd}. \end{cases}$$

Therefore, a sufficient condition of $\mathbb{E}\|\Phi_t - \nabla\varphi(\boldsymbol{x}_t)\| = \mathcal{O}(\epsilon)$ is

$$\begin{cases} \|\boldsymbol{y}_{t+1}^j - \boldsymbol{y}_{j\nu}^*(\boldsymbol{x}_t)\| = \mathcal{O}(\nu\epsilon/L_1), \ \forall j = -p/2, \cdots, p/2, & p \text{ is even}; \\[2mm] \|\boldsymbol{y}_{t+1}^j - \boldsymbol{y}_{j\nu}^*(\boldsymbol{x}_t)\| = \mathcal{O}(\nu\epsilon/L_1), \ \forall j = -(p-1)/2, \cdots, (p+1)/2, & p \text{ is odd}. \end{cases} \tag{18}$$

Our next goal is to show that our parameter setting fulfills Eq. (18). Note that for $\nu = \mathcal{O}(1/\kappa)$, the (perturbed) lower-level problem $g_{j\nu}(\boldsymbol{x}, \boldsymbol{y})$ is $\Omega(\mu)$-strongly convex in $\boldsymbol{y}$ and has $\mathcal{O}(L_1)$-Lipschitz continuous gradients jointly in $(\boldsymbol{x}, \boldsymbol{y})$. Therefore, if we set $\eta_y \lesssim 1/L_1$, then we can apply Lemma D.2 on the lower-level problem $g_{j\nu}(\boldsymbol{x}, \boldsymbol{y})$ to conclude that for ant $j$, we have

$$\mathbb{E}\|\boldsymbol{y}_{t+1} - \boldsymbol{y}_{j\nu}^*(\boldsymbol{x}_t)\|^2 \leq (1 - \mu\eta_y)^K \|\boldsymbol{y}_t - \boldsymbol{y}_{j\nu}^*(\boldsymbol{x}_t)\|^2 + \mathcal{O}(\eta_y\sigma^2/\mu).$$

Comparing it with Eq. (18), we can set $\eta_y = \mathcal{O}(\nu^2\epsilon^2/L_1\kappa\sigma^2)$ to ensure that for ant $j$, we have

$$\mathbb{E}\|\boldsymbol{y}_{t+1} - \boldsymbol{y}_{j\nu}^*(\boldsymbol{x}_t)\| \leq (1 - \mu\eta_y)^K \|\boldsymbol{y}_t - \boldsymbol{y}_{j\nu}^*(\boldsymbol{x}_t)\| + \mathcal{O}(\nu\epsilon/L_1).$$

Further, we can use Lemma D.4 and the triangle inequality to obtain that for ant $j$, we have

$$\mathbb{E}\|\boldsymbol{y}_{t+1} - \boldsymbol{y}_{j\nu}^*(\boldsymbol{x}_t)\| \leq (1 - \mu\eta_y)^K (\|\boldsymbol{y}_t - \boldsymbol{y}_{j\nu}^*(\boldsymbol{x}_{t-1})\| + 4\kappa\|\boldsymbol{x}_t - \boldsymbol{x}_{t-1}\|) + \mathcal{O}(\nu\epsilon/L_1). \quad (19)$$

The recursion (19) implies our setting of $K$ can ensure that Eq. (18) holds for all $t = 0, \cdots, T-1$. We give an induction-based proof. To let the induction base holds for $t = 1$, it suffices to set $K = \Omega(\log(RL_1/\nu\epsilon)/\mu\eta_y) = \Omega(\log(RL_1/\nu\epsilon)\kappa^2\sigma^2/\nu^2\epsilon^2)$, where $\|\boldsymbol{y}_{j\nu}^*(\boldsymbol{x}_0) - \boldsymbol{y}^*(\boldsymbol{x}_0)\|^2 = \mathcal{O}(R)$ is due to the setting of $\nu = \mathcal{O}(R/\kappa)$ and the fact that $\boldsymbol{y}_\nu^*(\boldsymbol{x})$ is $\kappa$-Lipschitz in $\nu$ by Lemma C.2. Next, assume that we have already guaranteed Eq. (18) for iteration $t$, we prove that our setting of $K$ implies Eq. (18) holds for iteration $t+1$. Note that the NSGD update in $\boldsymbol{x}$ means that $\|\boldsymbol{x}_t - \boldsymbol{x}_{t-1}\| = \eta_x = \mathcal{O}(\epsilon/6L_1\kappa^3)$. Therefore, Eq. (19) in conjunction with the induction hypothesis indicates that

$$\mathbb{E}\|\boldsymbol{y}_{t+1} - \boldsymbol{y}_{j\nu}^*(\boldsymbol{x}_t)\| \lesssim (1 - \mu\eta_y)^K \left( \frac{\nu\epsilon}{L_1} + \frac{\epsilon}{L_1\kappa^2} \right) + \frac{\nu\epsilon}{L_1}.$$

Therefore, we know that to let Eq. (18) holds for iteration $t+1$, it suffices to let $K = \Omega(\log(1/\nu\kappa^2)/\mu\eta_y) = \Omega(\log(1/\nu\kappa^2)\kappa^2\sigma^2/\nu^2\epsilon^2)$. This finishes the induction.

**Total Complexity.** According to the above analysis, we set $\nu \asymp (\epsilon/\bar{L}\kappa^{2p+1})^{1/p}$, $S \asymp \sigma^2/\nu^2\epsilon^2$, $T \asymp \Delta L_1\kappa^3\epsilon^{-2}$, and $K \asymp \log(RL_1\kappa/\nu\epsilon)\kappa^2\sigma^2/\nu^2\epsilon^2$ to ensure that the algorithm provably find an $\epsilon$-stationary point of $\varphi(\boldsymbol{x})$. Since $S \lesssim K$, the total complexity of the algorithm is

$$pT(S + K) = \mathcal{O}(pTK) = \mathcal{O}\left( p \cdot \frac{\Delta L_1\kappa^3}{\epsilon^2} \cdot \frac{\kappa^2\sigma^2}{\nu^2\epsilon^2} \log\left( \frac{RL_1\kappa}{\nu\epsilon} \right) \right)$$

$$= \mathcal{O}\left( \frac{p\Delta L_1\bar{L}^{2/p}\sigma^2\kappa^{9+2/p}}{\epsilon^{4+2/p}} \log\left( \frac{RL_1\kappa}{\nu\epsilon} \right) \right).$$

$\square$

# E    PROOF OF THEOREM 4.1

We prove our lower bound for stochastic nonconvex-strongly-convex bilevel optimization via a reduction to the lower bound for stochastic single-level nonconvex optimization (Arjevani et al., 2023). To state their lower bound, we first need to introduce the function class, oracle class, algorithm class, and the complexity measures.

**Definition E.1.** *Given any $\Delta > 0$ and $L_1 > 0$, we use $\mathcal{F}^{\mathrm{nc}}(L_1, \Delta)$ to denote the set of all smooth functions $f : \mathbb{R}^d \to \mathbb{R}$ that satisfies*

*1. $f(\mathbf{0}) - \inf_{\boldsymbol{x}\in\mathbb{R}^d} f(\boldsymbol{x}) \leq \Delta$;*

*2. $\nabla f(\boldsymbol{x})$ is $L_1$-Lipschitz continuous.*

**Definition E.2.** *Given a function $\mathbb{R}^d \to \mathbb{R}$, we use $\mathbb{O}(\sigma^2)$ to denote the set of all stochastic first-order oracles that return an unbiased stochastic estimator to $\nabla f$ with variance bounded by $\sigma^2$.*

**Definition E.3.** *Let $f : \mathbb{R}^d \to \mathbb{R}$ be a differentiable function and $F : \mathbb{R}^d \to \mathbb{R}$ be the stochastic estimator to $\nabla f$. A randomized first-order algorithm $\mathbb{A}$ consists of a distribution $\mathcal{P}_r$ over a measurable set $\mathcal{R}$ and a sequence of measurable mappings $\{\mathbb{A}_t\}_{t\in\mathbb{N}}$ such that*

$$\boldsymbol{x}_{t+1} = \mathbb{A}_t(r, F(\boldsymbol{x}_0), \cdots, F(\boldsymbol{x}_t)), \quad t \in \mathbb{N}_+,$$

*where $r \sim \mathcal{P}_r$ is drawn a single time at the beginning of the protocol. We let $\mathcal{A}_{\mathrm{rand}}$ to denote the class of all the algorithms that follow the above protocol.*

**Definition E.4.** *We define distributional complexity of $\mathcal{A}_{\mathrm{rand}}$ to find an $\epsilon$-stationary point of the functions in $\mathcal{F}^{\mathrm{nc}}(L_1, \Delta)$ with oracle $\mathbb{O}(\sigma^2)$ as*

$$\mathrm{Compl}_\epsilon(L_1, \Delta, \sigma^2) = \sup_{\mathcal{O}\in\mathbb{O}(\sigma^2)} \sup_{\mathcal{P}_f\in\mathcal{P}[\mathcal{F}(\Delta, f)]} \inf_{\mathbb{A}\in\mathcal{A}_{\mathrm{rand}}} \inf\{t \in \mathbb{N} \mid \mathbb{E}\|\nabla f(\boldsymbol{x}_t)\| \leq \epsilon\},$$

*where the expectation is taken over the sampling of $f$ from $\mathcal{P}_f$, the randomness in the oracle $\mathcal{O}$, and the randomness in the algorithm $\mathbb{A}$, $\{\boldsymbol{x}_t\}_{t\in\mathbb{N}}$ is the sequence generated by $\mathbb{A}$ running on $f$ with oracle $\mathcal{O}$, and $\mathcal{P}[\mathcal{F}^{\mathrm{nc}}(L_1, \Delta)]$ denotes the set of all distributions over $\mathcal{F}^{\mathrm{nc}}(L_1, \Delta)$.*

All the above definitions are merely restatements of (Arjevani et al., 2023, Section 2). Although Definition E.4 uses the definition of distributional complexity, by Yao's minimax principle is also a lower bound for the worst-case complexity. Now, we recall the construction in (Arjevani et al., 2023) for proving the $\Omega(\epsilon^{-4})$ lower bound. Formally, we define the randomized function

$$f_{\boldsymbol{U}}(\boldsymbol{x}) = \frac{L_1 \beta^2}{\bar{L}_1} f^{\mathrm{nc}}(\rho(\boldsymbol{U}^\top \boldsymbol{x}/\beta)) + \frac{L_1 \lambda}{2\bar{L}_1} \|\boldsymbol{x}\|^2, \tag{20}$$

where $\bar{L}_1 = 155$, $\beta = 4\bar{L}_1\epsilon/L_1$, $\rho : \mathbb{R}^T \to \mathbb{R}^T$ is $\rho(\boldsymbol{x}) = \boldsymbol{x}/\sqrt{1 + \|\boldsymbol{x}\|^2/R^2}$, $R = 230\sqrt{T}$, $\lambda = 1/5$, and $f_T : \mathbb{R}^T \to \mathbb{R}$ is the nonconvex hard instance introduced by Carmon et al. (2020):

$$f^{\mathrm{nc}}(x) := -\Psi(1)\Psi(x_1) + \sum_{i=2}^{T} [\Phi(-x_{i-1})\Phi(-x_i) - \Psi(x_{i-1})\Psi(x_i)].$$

In the above, the component functions $\Psi, \Phi : \mathbb{R} \to \mathbb{R}$ are defined as

$$\Psi(t) = \begin{cases} 0, & t \le 1/2, \\ \exp(1 - 1/(2t-1)^2), & t < 1/2 \end{cases} \quad \text{and} \quad \Phi(t) = \sqrt{e} \int_{-\infty}^{t} \exp(-t^2/2)\mathrm{d}t.$$

For the hard instance in Eq. (20), Arjevani et al. (2023) further defined the stochastic gradient estimator $F_{\boldsymbol{U}}$ as

$$F_{\boldsymbol{U}}(\boldsymbol{x}) = \frac{L_1}{\bar{L}_1} \left( \beta(\nabla\rho(\boldsymbol{x}))^\top \boldsymbol{U} F_T(\boldsymbol{U}^\top \rho(\boldsymbol{x})) + \lambda \boldsymbol{x} \right). \tag{21}$$

In the above, $F_T : \mathbb{R}^T \to \mathbb{R}^T$ is the stochastic gradient estimator of $\nabla f^{\mathrm{nc}}$ defined by

$$[F_T(\boldsymbol{x})]_i = \nabla_i f^{\mathrm{nc}}(\boldsymbol{x}) \left( 1 + \mathbf{1}_{i > \mathrm{prog}_{1/4}(x)} (\xi/\gamma - 1) \right), \quad \xi \sim \mathrm{Bernoulli}(\gamma),$$

where $\mathrm{prog}_\alpha(x) = \max\{i \ge 0 \mid |x_i| > \alpha\}$ and $\gamma = \min\{(46\epsilon)^2/\sigma^2, 1\}$. For the above construction, Arjevani et al. (2023) showed the following lower bound.

**Theorem E.1** ((Arjevani et al., 2023, Theorem 3)). *There exist numerical constants $c, c' > 0$ such that for all $\Delta > 0$, $L_1 > 0$ and $\epsilon \le c\sqrt{L_1\Delta}$, the construction of function $f_{\boldsymbol{U}} : \mathbb{R}^d \to \mathbb{R}$ and stochastic first-order oracle $F_{\boldsymbol{U}} : \mathbb{R}^d \to \mathbb{R}$ in Eq. (20) and (21) together give a distribution over the function class $\mathcal{F}^{\mathrm{nc}}(L_1, \Delta)$ and a stochastic first-order oracle $\mathbb{O} \in \mathbb{O}(\sigma^2)$ such that*

$$\mathrm{Compl}_\epsilon(L_1, \Delta, \sigma^2) \ge c'\Delta L_1 \sigma^2 \epsilon^{-4}.$$

*Proof of Theorem 4.1.* For any randomized algorithm $\mathtt{A}$ defined as Eq. (11) running it on our hard instance, we show that it can be simulated by another randomized algorithm running on the variable $\boldsymbol{x}$ such that Theorem E.1 can be applied. Since $G(y) = \mu y$ is a deterministic mapping we know that any randomized algorithm $\mathtt{A}$ induces a sequence of measurable mappings $\{\mathtt{A}'_t\}_{t \in \mathbb{N}}$ such that

$$(\boldsymbol{x}_t, y_t) = \mathtt{A}'_t(\xi, F(\boldsymbol{x}_0), \cdots, F(\boldsymbol{x}_{t-1}), y_0, \cdots, y_{t-1}).$$

Expanding the recursion for $y_t$ shows that the above equation induces another sequence of measurable mappings $\{\mathtt{A}''_t\}_{t \in \mathbb{N}}$ such that

$$(\boldsymbol{x}_t, y_t) = \mathtt{A}''_t(\xi, F(\boldsymbol{x}_0), \cdots, F(\boldsymbol{x}_{t-1})).$$

Therefore, we can apply Theorem E.1 to complete the proof. $\qquad \square$

# F  THE F$^2$SA-2 ALGORITHM

We present the realization of F$^2$SA-$p$ when $p = 2$ in Algorithm 3 to further compare its procedure with the original F$^2$SA algorithm. Let $\lambda = 1/\nu$. We can observe that F$^2$SA (Kwon et al., 2023; Chen et al., 2025b) solves the following *asymmetric* penalty problem

$$\min_{\boldsymbol{x} \in \mathbb{R}^{d_x}, \boldsymbol{y} \in \mathbb{R}^{d_y}} f(\boldsymbol{x}, \boldsymbol{y}) + \lambda \left( g(\boldsymbol{x}, \boldsymbol{y}) - \min_{\boldsymbol{z} \in \mathbb{R}^{d_y}} g(\boldsymbol{x}, \boldsymbol{z}) \right),$$

As an improved method, the F$^2$SA-2 solved the following *symmetric* penalty problem:

$$\min_{\boldsymbol{x} \in \mathbb{R}^{d_x}, \boldsymbol{y} \in \mathbb{R}^{d_y}} \frac{1}{2} \left( f(\boldsymbol{x}, \boldsymbol{y}) + \lambda f(\boldsymbol{x}, \boldsymbol{y}) - \min_{\boldsymbol{z} \in \mathbb{R}^{d_y}} (-f(\boldsymbol{x}, \boldsymbol{z}) + \lambda g(\boldsymbol{x}, \boldsymbol{z})) \right).$$

The latter is better since the symmetric form makes the first-order approximation error to $\nabla\varphi(\boldsymbol{x})$ perfectly cancel out and leave only the second-order error term. Therefore, in terms of the theoretical guarantee by Theorem 3.1, the $\tilde{\mathcal{O}}(\epsilon^{-5})$ upper bound of F$^2$SA-2 can improve the $\tilde{\mathcal{O}}(\epsilon^{-6})$ upper bound of F$^2$SA by a factor of $\epsilon^{-1}$.

---

**Algorithm 3** F$^2$SA-2 $(\boldsymbol{x}_0, \boldsymbol{y}_0)$

---

1: $\boldsymbol{z}_0 = \boldsymbol{y}_0$
2: **for** $t = 0, 1, \cdots, T-1$
3: $\quad$ $\boldsymbol{y}_t^0 = \boldsymbol{y}_t$, $\boldsymbol{z}_t^0 = \boldsymbol{z}_t$
4: $\quad$ **for** $k = 0, 1, \cdots, K-1$
5: $\qquad$ Sample random i.i.d indexes $(\xi^y, \zeta^y)$ and $(\xi^z, \zeta^z)$.
6: $\qquad$ $\boldsymbol{y}_t^{k+1} = \boldsymbol{y}_t^k - \eta_y \left( \nu F_y(\boldsymbol{x}_t, \boldsymbol{y}_t^k; \xi^y) + G_y(\boldsymbol{x}_t, \boldsymbol{y}_t^k; \zeta^y) \right)$
7: $\qquad$ $\boldsymbol{z}_t^{k+1} = \boldsymbol{z}_t^k - \eta_y \left( -\nu F_y(\boldsymbol{x}_t, \boldsymbol{z}_t^t; \xi^z) + G_y(\boldsymbol{x}_t, \boldsymbol{z}_t^k; \zeta^z) \right)$
8: $\quad$ **end for**
9: $\quad$ $\boldsymbol{y}_{t+1} = \boldsymbol{y}_t^K$, $\boldsymbol{z}_{t+1} = \boldsymbol{z}_t^K$
10: $\quad$ Sample random i.i.d indexes $\{(\xi_i^x, \zeta_i^x)\}_{i=1}^S$.
11: $\quad$ $\Phi_t = \frac{1}{2} \sum_{i=1}^S \left( F_x(\boldsymbol{x}_t, \boldsymbol{y}_{t+1}; \xi_i^x) + F_x(\boldsymbol{x}_t, \boldsymbol{z}_{t+1}; \xi_i^x) + \frac{G_x(\boldsymbol{x}_t, \boldsymbol{y}_{t+1}; \zeta_i^x) - G_x(\boldsymbol{x}_t, \boldsymbol{z}_{t+1}; \zeta_i^x)}{\nu} \right)$
12: $\quad$ $\boldsymbol{x}_{t+1} = \boldsymbol{x}_t - \eta_x \Phi_t / \|\Phi_t\|$
13: **end for**

---

## G  ADDITIONAL EXPERIMENTS

This section provides additional experiments on finding the optimal per-parameter regularization of a 5-layer MLP with ReLU activation and the hidden layer size of $500$. Following the notation in Example 2.2, we let $\boldsymbol{x} \in \mathbb{R}^d$ parameterize the regularization matrix via $\boldsymbol{W_x} = \mathrm{diag}(\exp(\boldsymbol{x}))$. We also let $\ell_{\mathrm{val}}$ and $\ell_{\mathrm{tr}}$ be the logistic loss of the network prediction on the validation set and training set, respectively. The problem to solve has the same formulation as Example 2.2, as restated below:

$$\min_{\boldsymbol{x} \in \mathbb{R}^d} \ell_{\mathrm{val}}(\boldsymbol{y}), \quad \text{s.t.} \quad \boldsymbol{y} \in \arg \min_{\boldsymbol{y} \in \mathbb{R}^d} \ell_{\mathrm{tr}}(\boldsymbol{y}) + \boldsymbol{y}^\top \boldsymbol{W_x} \boldsymbol{y}. \tag{22}$$

The difference between Example 2.2 is that now the problem is nonsmooth nonconvex due to the use of the MLP model. We present the experiment results in Figure 2.

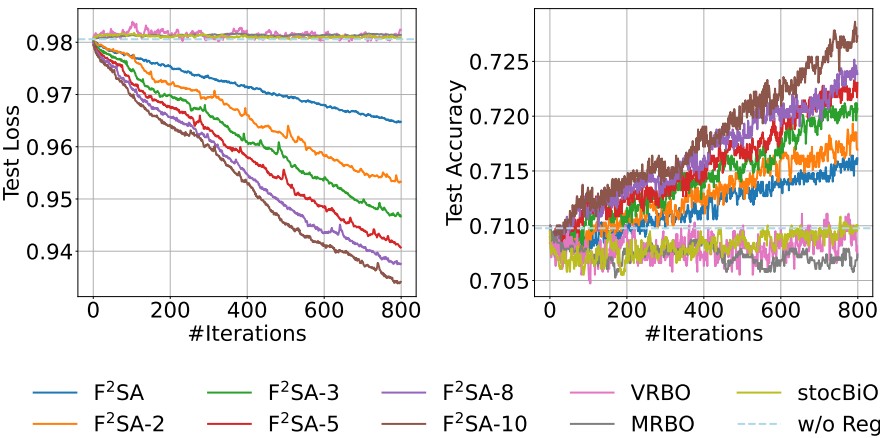

Figure 2: Performances of different algorithms on Problem (22) with an MLP model.

## H  USE OF LARGE LANGUAGE MODELS

Large language models were used to help calculate the coefficient $\alpha_0$ when $p$ is odd in Lemma 3.1, and to refine wording and correct grammatical errors in parts of the paper.