# OpenReview forum: "Faster Gradient Methods for Highly-smooth Stochastic Bilevel Optimization"
_ICLR.cc/2026/Conference — ICLR 2026 Poster_

### Official Review · Reviewer_mbBx · 2025-10-28

**Soundness:** 3
**Presentation:** 3
**Contribution:** 4
**Rating:** 8
**Confidence:** 4

**Summary:**

This paper provides a conceptually elegant and mathematically rigorous advancement in stochastic bilevel optimization. By linking finite-difference theory with $p$th-order first-order hyper-gradient estimation, it achieves improved convergence without Hessian information. The analysis is solid and the experiments validate both theory and practicality, though assumptions on high-order smoothness and complexity constants limit immediate applicability to all real-world problems. Nonetheless, it represents a notable theoretical milestone toward closing the gap between first-order bilevel optimization and its single-level counterpart.

**Strengths:**

This paper provides a novel and elegant finite-difference interpretation of first-order bilevel optimization, enabling the F2SA-p method to achieve near-optimal convergence without any Hessian computation. It successfully bridges theory and practice by offering a fully first-order, scalable, and provably faster framework for highly smooth stochastic bilevel problems.

**Weaknesses:**

This paper relies on a new assumption of high-order smoothness in the lower-level variable $y$. The improved convergence rate largely stems from this assumption; however, since the method requires a relatively large $p$, the assumption may be quite strong in practice.

**Questions:**

Could the authors clarify how much Assumption 2.5 (high-order smoothness) contributes to the improvement in the convergence rate? Additionally, could the authors provide justification or empirical evidence regarding the strength or practicality of this assumption?

---

> ### Author Response · Authors · 2025-11-20
>
> We greatly appreciate the reviewer's support of our work, and we respond to his/her questions one by one below.
>
> ## Weaknesses
>
> > W1. This paper relies on a new assumption of high-order smoothness in the lower-level variable. The improved convergence rate largely stems from this assumption; however, since the method requires a relatively large $p$, the assumption may be quite strong in practice.
>
> We clarify that our method can work for any $p \in \mathbb{N}_+$ and does not necessarily "require a relatively large $p$.". The additional smoothness for $p=2$ is standard for accelerated nonconvex optimization [4-6]. For applications that satisfy the smoothness for $p \ge 3$, please refer to our response to Q2 below.
>
> ## Question
>
> > Q1. Could the authors clarify how much Assumption 2.5 (high-order smoothness) contributes to the improvement in the convergence rate?
>
> For $p$th order smooth problems, the F${}^2$SA-$p$ method achieves the complexity of $\tilde{\mathcal{O}}(\epsilon^{-4-2/p})$. Smooth problems have larger $p$ and hence faster convergence rates.
>
> > Q2. Additionally, could the authors provide justification or empirical evidence regarding the strength or practicality of this assumption?
>
> A classical example of a highly smooth function is the softmax function [Lemma 2(3), 1]. Therefore, many hyper-parameter tuning problems for logistic regression are provably highly smooth, including the problem in our experiment section (Section 5) and the data-hypercleaning problem in [2,3]. For their formal mathematical formulations, please refer to Example 2.1 and 2.2 in our revision.
>
> ## Reference in the Response
>
> [1]  Ankit Garg, Robin Kothari, Praneeth Netrapalli, and Suhail Sherif. _Near-optimal lower bounds for convex optimization for all orders of smoothness_. In NeurIPS, 2021.
>
> [2] Luca Franceschi, Paolo Frasconi, Saverio Salzo, Riccardo Grazzi, and Massimiliano Pontil. _Bilevel programming for hyperparameter optimization and meta-learning_. In ICML, 2018.
>
> [3] Kaiyi Ji, Junjie Yang, and Yingbin Liang. _Bilevel optimization: Convergence analysis and enhanced design_. In ICML, 2021.
>
> [4] Nilesh Tripuraneni, Mitchell Stern, Chi Jin, Jeffrey Regier, and Michael I. Jordan. _Stochastic cubic regularization for fast nonconvex optimization_. In NeurIPS, 2018.
>
> [5] Cong Fang, Zhouchen Lin, and Tong Zhang. _Sharp analysis for nonconvex SGD escaping from saddle points_. In COLT, 2019.
>
> [6] Ashok Cutkosky and Harsh Mehta. _Momentum improves normalized SGD_. In ICML, 2020.

---

### Official Review · Reviewer_pQay · 2025-10-30

**Soundness:** 3
**Presentation:** 3
**Contribution:** 3
**Rating:** 6
**Confidence:** 3

**Summary:**

This paper studies the stochastic bilevel optimization problem in which the upper-level objective is nonconvex and the lower-level objective is strongly convex. Building on the first-order F2SA algorithm, which achieves $\tilde{O}(\epsilon^{-6})$ stochastic oracle complexity, the authors propose a family of methods F2SA-p that leverage p-th order finite-difference hyper-gradient approximation. They prove that under p-th order smoothness in the lower-level variable, the algorithm attains a tighter complexity bound $\tilde{O}(p \kappa^{9+2/p} \epsilon^{-4 - 2/p})$. Moreover, they establish a matching lower bound $\Omega(\epsilon^{-4})$ under similar smoothness assumptions, showing near-optimality when $p = \Omega(\log(1/\epsilon)/\log\log(1/\epsilon))$.

**Strengths:**

1. The finite-difference modification of F2SA is both novel and conceptually elegant, offering a unified view that also tightens and refines previous analyses.
2. The paper provides an explicit comparison of smoothness assumptions with related works and includes a thorough and contextualized literature review.
3. A new lower-bound result demonstrates that the upper bound is nearly optimal in the regime $p = \Omega(\log(1/\epsilon)/\log\log(1/\epsilon))$.

**Weaknesses:**

1. The experiments are limited to a single small-scale dataset, with no evaluation on modern large-scale or deep bilevel tasks (e.g., hyperparameter tuning or meta-learning).
2. The high-order smoothness assumption on the lower-level variable may be restrictive or difficult to verify in practical ML settings. It would strengthen the paper to discuss or cite works examining when such smoothness assumptions hold in real-world problems.

**Questions:**

1. Regarding the convergence guarantee in Theorem 3.1, is the result stated in expectation or with high probability?
2. I wonder whether the proposed algorithm can be combined with variance-reduction or momentum techniques to further improve the convergence rate.

---

> ### Author Response · Authors · 2025-11-20
>
> We greatly thank the reviewer for taking his/her time to review the manuscript and providing constructive feedback. We respond to his/her comments one by one below.
>
> ## Weaknesses
>
> > W1. The experiments are limited to a single small-scale dataset, with no evaluation on modern large-scale or deep bilevel tasks (e.g., hyperparameter tuning or meta-learning).
>
> We greatly appreciate your suggestions for enhancing our experiments. We have provided extra experiments on tuning the regularization of an MLP model in Appendix G.
>
> > W2. The high-order smoothness assumption on the lower-level variable may be restrictive or difficult to verify in practical ML settings. It would strengthen the paper to discuss or cite works examining when such smoothness assumptions hold in real-world problems.
>
> A classical example of a highly smooth function is the softmax function [Lemma 2(3), 1]. Therefore, many hyper-parameter tuning problems for logistic regression are provably highly smooth, including the problem in our experiment section (Section 5) and the data-hypercleaning problem in [2,3]. For their formal mathematical formulations, please refer to Example 2.1 and 2.2 in our revision.
>
> ## Questions
>
> > Q1. Regarding the convergence guarantee in Theorem 3.1, is the result stated in expectation or with high probability?
>
> The result in Theorem 3.1 is stated in expectation. Thanks a lot for pointing out the potential ambiguity. We have made modifications in Definition 2.1, which defines an $\epsilon$-stationary point as $\mathbb{E} \Vert \nabla \varphi(x) \Vert \le \epsilon$.
>
> > Q2. I wonder whether the proposed algorithm can be combined with variance-reduction or momentum techniques to further improve the convergence rate.
>
> We think it is quite possible to combine with variance reduction techniques, such as those proposed in [4,5], to achieve a complexity of $\tilde{\mathcal{O}}(\epsilon^{-3-p/2})$ under the mean-squared smoothness assumption. We leave the study of variance-reduced methods for future work.
>
> ## Reference in the Response
>
> [1] Ankit Garg, Robin Kothari, Praneeth Netrapalli, and Suhail Sherif. _Near-optimal lower bounds for convex optimization for all orders of smoothness_. In NeurIPS, 2021.
>
> [2]  Luca Franceschi, Paolo Frasconi, Saverio Salzo, Riccardo Grazzi, and Massimiliano Pontil. _Bilevel programming for hyperparameter optimization and meta-learning_. In ICML, 2018.
>
> [3] Kaiyi Ji , Junjie Yang, and Yingbin Liang. _Bilevel optimization: Convergence analysis and enhanced design_. In ICML, 2021.
>
> [4] Cong Fang, Chris Junchi Li, Zhouchen Lin, and Tong Zhang. _SPIDER: Near-optimal non-convex optimization via stochastic path-integrated differential estimator_. In NeurIPS, 2018.
>
> [5] Ashok Cutkosky and Francesco Orabona. _Momentum-based variance reduction in non-convex SGD_. In NeurIPS, 2019.

---

### Official Review · Reviewer_MZzi · 2025-11-04

**Soundness:** 3
**Presentation:** 2
**Contribution:** 3
**Rating:** 6
**Confidence:** 3

**Summary:**

In this paper, the authors study a stochastic bilevel optimization problem where the upper-level objective is non-convex and the lower-level objective is strongly convex. Instead of relying on Hessian-vector products, they focus on first-order methods that use only gradient information. The key insight is that the state-of-the-art first-order method, $\textsf{F}^2\textsf{SA}$, can be viewed as approximating the hypergradient via a forward difference of a univariate function. Building on this idea, the authors employ higher-order finite difference schemes and, under proper smoothness assumptions, establish a complexity bound of $\tilde{O}(p \kappa^{9+2/p} \epsilon^{-4-2/p})$. When $p=1$, their result improves the previous analysis by a factor of $\kappa$, and when $p = \Omega(\log \epsilon^{-1} / \log\log \epsilon^{-1})$, the complexity improves to $\tilde{O}(\kappa^9 \epsilon^{-4})$, matching the optimal rate for single level problems.

**Strengths:**

- The $\textsf{F}^2\textsf{SA}$ method was originally proposed as a penalty-based approach. In this paper, the authors offer a novel interpretation by viewing it as a forward difference approximation of the hypergradient. This perspective naturally extends to higher-order finite-difference schemes and provides a clear pathway to improve first-order methods.
- The authors also perform a tight analysis to bound the Lipschitz constants of the higher-order derivatives of the perturbed function in terms of those of the original objectives. Under additional smoothness assumptions, they derive a sharper complexity bound than existing first-order methods.

**Weaknesses:**

- **Dimension dependence in Remark 3.1**. The remark states that Lemma 3.1 can be applied to each dimension and thus the overall error can be controlled in Euclidean norm. However, I think a direct coordinate-wise argument would incur an extra $\sqrt{d}$ factor in the approximation error, which could be problematic. Could you please clarify whether your analysis avoids this factor, e.g., by analyzing the error in different coordinates jointly?
- **Lower bound in Section 4**. The construction effectively considers a bilevel problem where the upper-level and lower-level objectives are independent, thus reducing to a single-level non-convex problem. While I am not aware of a prior formal statement in exactly this form, the resulting conclusion feels fairly direct.
- The presentation can be improved, and several points would benefit from additional detail:
    - In the context of the penalty-based method, $\nu$ plays the role of the inverse of the penalty parameter $\lambda$, so I find it somewhat counterintuitive that the algorithm considers negative $\nu$ when $p$ is even. This also raises the question whether $\min_{y} g_{\nu}(x,y)$ can be solved efficiently. My understanding is that taking $\nu$ sufficiently small preserves smoothness and strong convexity in $y$, so the subproblem can still be solved in a standard way.
   - On a related note, the relationship between the symmetric penalty problem in (4), the penalty problem in (3), and the original bilevel objective is not immediately clear to me. A brief discussion clarifying their connections will be useful.
    - Typos:
         - Page 5, Line 262. The identity $\frac{\partial}{\partial \nu} \ell_{\nu}(x) \vert_{\nu=0} = \phi(x)$ is cited to Chen et al. (2025b), Lemma B.3. However, that lemma appears to bound the gap between the penalized function and the hyperobjective.
         - In Lemma 3.1, the coefficients for even $p$ should satisify $\alpha_j = -\alpha_{-j}$. Also, for the $p$th-order forward difference estimator, the indices should run $j=0, \dots, p$.
         - In Lemma D.2, the SGD update is written as the normalized SGD by mistake.

**Questions:**

Please see the "Weaknesses" Section above.

---

> ### Author Response · Authors · 2025-11-20
>
> We greatly thank the reviewer for taking his/her time to review the manuscript and providing constructive feedback. We respond to his/her comments one by one below.
>
> ## Weakness / Question
>
> > 1. **Dimension dependence in Remark 3.1.** I think a direct coordinate-wise argument would incur an extra $\sqrt{d}$ factor in the approximation error, which could be problematic.
>
> Thanks a lot for pointing this out! We agree that a direct coordinate-wise argument would incur an extra dimension factor in the approximation error. We apologize for any confusion caused by our Remark 3.1.
>
> We have revised the statement in Lemma 3.1 to apply to vector-valued functions $\psi: \mathbb{R}^d \rightarrow \mathbb{R}$ and removed Remark 3.1. As you conjectured, our analysis avoids this factor by jointly analyzing the error in different coordinates. Notably, the proof is the same as the unitary case $\psi: \mathbb{R} \rightarrow \mathbb{R}$ since the potential dimension dependency is already **hidden** in the Lipschitz condition $\Vert \psi^{(p)}(\nu) -  \psi^{(p)}(\nu') \Vert \le C \Vert \nu - \nu' \Vert$.
>
>
> > 2. **Lower bound in Section 4.** The construction effectively considers a bilevel problem where the upper-level and lower-level objectives are independent, thus reducing to a single-level non-convex problem. While I am not aware of a prior formal statement in exactly this form, the resulting conclusion feels fairly direct.
>
> The main purpose of Section 4 is to show that the $\Omega(\epsilon^{-4})$ lower bound also holds for highly-smooth bilevel problems in Definition 2.2 and for the algorithm class defined in Eq. (11).
>
> Intuitively, bilevel problems should be harder than single-level problems, and we agree that the resulting conclusion may "feel fairly direct", as you said. However, we find that existing constructions [1,2] (or their direct extensions) do not satisfy the highly smooth assumption and can not be directly applied in our setting; therefore, we provide our construction for completeness.
>
> ## Reference in the Response
>
> [1] Mathieu Dagr´eou, Thomas Moreau, Samuel Vaiter, and Pierre Ablin. _A lower bound and a near-optimal algorithm for bilevel empirical risk minimization_. In AISTATS, 2024.
>
> [2] Jeongyeol Kwon, Dohyun Kwon, and Hanbaek Lyu. _On the complexity of first-order methods in
> stochastic bilevel optimization_. In ICML, 2024.

---

> ### Author Response · Authors · 2025-11-20
> **Further Response to the Presentation Feedback**
>
> We sincerely appreciate your careful review, which greatly helps us improve the presentation of our manuscript. We have revised our manuscript according to your suggestions (revisions are marked in blue). In the following, we provide explanations regarding your questions.
>
> > 1. In the context of the penalty-based method, $\nu$ plays the role of the inverse of the penalty parameter $\lambda$, so I find it somewhat counterintuitive that the algorithm considers negative $\nu$ when is $p$ even. This also raises the question whether $\min_y g_\nu(x,y)$ can be solved efficiently. My understanding is that taking sufficiently small $\nu$ preserves smoothness and strong convexity in $g_\nu(x,y)$, so the subproblem can still be solved in a standard way.
>
> For $p=2$, the symmetric penalty problem in (4) is
>
> $$
> \min_x \frac{1}{2} \left( f(x,y_{\lambda}^\ast(x)) + f(x,z_{\lambda}^\ast(x)) + \lambda (g(x,y_{\lambda}^\ast(x)) - g(x,z_{\lambda}^\ast(x))) \right),
> $$
>
> subject to
>
> $$
> \begin{cases}
> y_{\lambda}^\ast(x) = \arg \min_{y} \{f(x,y) + \lambda g(x,y) \} \\\\ z_{\lambda}^\ast(x) = \arg \min_{z} \{-f(x,z) + \lambda g(x,z)\}.
> \end{cases}
> $$
> It is still a penalty problem with positive $\lambda$, except $y,z$ are perturbed in the opposite direction to better cancel out the errors.
>
> As for solving the subproblem $g_\nu(x,y)$, your understanding is exactly correct. Formally, we have that $\nabla_{yy}^2 g(x,y) \succeq -L_1 I$ since $f$ is $L_1$-smooth, and that $\nabla_{yy}^2 g(x,y) \succeq \mu_y I$ since $g$ is $\mu_y$ strongly convex in $y$. Therefore, $g_\nu(x,y) = \nu f(x,y) + g(x,y)$ is $\mu_y/2$-strongly convex in $y$ when $\vert \nu \vert \le \mu_y / (2L_1)$, meaning that it can still be solved in a standard way as you said.
>
> > 2. On a related note, the relationship between the symmetric penalty problem in (4), the penalty problem in (3), and the original bilevel objective is not immediately clear to me. A brief discussion clarifying their connections will be useful.
>
> The original problem is $\min_{x,y} f(x,y)$ subject to $y^\ast(x) = \arg \min_{y} g(x,y)$, whose constraint is equivalent to $g(x,y) \le g(x,y^\ast(x))$. Consequently, the Lagrangian function with multiplier $\lambda$ of this constrained problem is $\min_{x,y}f(x,y) + \lambda (g(x,y) - g(x,y^\ast(x))$, which corresponds to Problem (3). The relationship between Problem (4) and the original problem can be understood in a similar manner.
>
> > 3. Page 5, Line 262. The identity $\frac{\partial}{\partial \nu} \ell_\nu(x)\mid_{\nu = 0} = \varphi(x)$ is cited to Chen et al. (2025b), Lemma B.3. However, that lemma appears to bound the gap between the penalized function and the hyperobjective.
>
> The cited lemma, with $\lambda = 1/ \nu$, states that
>
> $$
> \left(\min_y f(x,y) + \frac{g(x,y) - g(x,y^*(x))}{\nu} \right) - \varphi(x) = O(\nu).
> $$
>
> We can show that it implies the identity $\frac{\partial}{\partial \nu} \ell_\nu(x)\mid_{\nu = 0} = \varphi(x)$ through simple calculations:
>
> $$
> \frac{\partial}{\partial \nu} \ell_\nu(x)\mid_{\nu = 0} = \lim_{\nu \rightarrow 0} \frac{\ell_\nu(x) - \ell_0(x)}{\nu} = \lim_{\nu \rightarrow 0}  \left(\min_y f(x,y) + \frac{g(x,y) - g(x,y^\ast(x))}{\nu} \right) =\varphi(x),
> $$
>
> where the second equality uses $\ell_\nu(x,y) = \min_y \{ \nu f(x,y) + g(x,y)\}$, and the last one uses the cited lemma with $\nu \rightarrow 0$.
>
> > 4 Typos. In Lemma 3.1, the coefficients for even $p$ should satisify $\alpha_j = -\alpha_{-j}$. Also, for the $p$th-order forward difference estimator, the indices should run $j =0,\cdots,p$. In Lemma D.2, the SGD update is written as the normalized SGD by mistake.
>
> Thank you for pointing them out! We have corrected them in our revision.

---

### Official Review · Reviewer_Rwzc · 2025-11-04

**Soundness:** 2
**Presentation:** 2
**Contribution:** 2
**Rating:** 4
**Confidence:** 3

**Summary:**

This paper proposes a variant of the first-order method, F2SA, which uses higher-order central/forward finite difference to approximate the hypergradient. Therefore, the proposed method is called ‌F2SA-p, which uses p points to construct an estimator to the derivative of a unitary function (e.g., p = 1 or 2).

**Strengths:**

Strength:

1. The work is ‌well-motivated‌, and provides both upper and lower bound analysis.

2. For ‌pth-order smooth bilevel problems‌, the authors sated that the proposed method achieves an improved lower bound.

**Weaknesses:**

Weakness:

1. What's the difference between the proposed method with F2SA in (Kwon et al., 2024a) and (Chen et al., 2025b)?

2. The ‌problem setup‌ in Section 2 aligns with prior works like Chen et al. (2025b) and Kwon et al. (2024a), with the addition of ‌Assumption 2.5‌ (high-order smoothness).

3. Fully first-order methods are ‌more scalable‌ than Hessian-vector-product-based approaches, particularly for large-scale problems like LLM training. What's the advantage of the proposed method with p= 2 or 3?

4. The experimental results are not convincing. For instance, the authors should report the performance of the proposed method when p= 3 or 4.

**Questions:**

1. What's the difference between the proposed method with F2SA in (Kwon et al., 2024a) and (Chen et al., 2025b)?

2. The ‌problem setup‌ in Section 2 aligns with prior works like Chen et al. (2025b) and Kwon et al. (2024a), with the addition of ‌Assumption 2.5‌ (high-order smoothness).

3. Fully first-order methods are ‌more scalable‌ than Hessian-vector-product-based approaches, particularly for large-scale problems like LLM training. What's the advantage of the proposed method with p= 2 or 3?

4. The experimental results are not convincing. For instance, the authors should report the performance of the proposed method when p= 3 or 4.

---

> ### Author Response · Authors · 2025-11-20
>
> We greatly thank the reviewer for taking his/her time to review the manuscript and providing constructive feedback. We respond to his/her comments one by one below.
>
> ## Weakness / Question
>
> > 1. What's the difference between the proposed method with F2SA in (Kwon et al., 2024a) and (Chen et al., 2025b)?
>
> F${}^2$SA [1,2] solves the penalty problem
>
> $$
> \min_{x,y} f(x,y) + \lambda (g(x,y) - \min_z g(x,z)).
> $$
>
> Our work demonstrates that, when the problem is highly smooth, faster rates can be achieved by solving the following generalized penalty problem:
>
> $$
> \begin{cases}
> \min_x \sum_{j=-p/2}^{p/2} \alpha_j \min_{y}  \left( j f(x,y) + \lambda g(x,y) \right), & \text{if } p \text{ is even}; \\\\
> \min_x \sum_{j=-(p-1)/2}^{(p+1)/2} \alpha_j \min_{y}  \left( j f(x,y) + \lambda g(x,y) \right), & \text{if } p \text{ is odd}.
> \end{cases}
> $$
> Here, $\\{\alpha_j\\}$ are the $p$th-order finite difference coefficients as defined in Lemma 3.1. For $p=1$, this formulation is equivalent to the penalty problem in [1,2] since $\alpha_0= -1$ and $\alpha_1 = 1$. However, for $p\ge 2$, our method solves a different penalty problem.
>
>
> > 2. The ‌problem setup‌ in Section 2 aligns with prior works like Chen et al. (2025b) and Kwon et al. (2024a), with the addition of ‌Assumption 2.5‌ (high-order smoothness).
>
> When $p=1$, Assumption 2.5 is subsumed by Assumption 2.4, which means
> our method can work under the same assumptions and recover the $\tilde{\mathcal{O}}(\epsilon^{-6})$ result in [1,2] in this scenario. When $p \ge 2$, our $\tilde{\mathcal{O}}(\epsilon^{-4-2/p})$ result is better than that of [1,2] by exploiting higher-order smoothness of the problem.
>
>
> > 3. Fully first-order methods are ‌more scalable‌ than Hessian-vector-product-based approaches, particularly for large-scale problems like LLM training. What's the advantage of the proposed method with $p= 2$ or $3$?
>
> Our algorithm is still fully first-order, and it does not use higher-order oracles (e.g. Hessian or Hessian-vector product). If the problem is $p$th-order smooth for $p \ge 2$, then applying F${}^2$SA-$p$ finds an $\epsilon$-stationary point in an improved complexity of $\tilde{\mathcal{O}}(\epsilon^{-4-2/p})$. For $p=2$, the $\tilde{\mathcal{O}}(\epsilon^{-5})$ complexity improves that of F${}^2$SA by a factor of $\epsilon^{-1}$. For $p = 3$, the $\tilde{\mathcal{O}}(\epsilon^{-14/3})$ complexity improves that of F${}^2$SA by a factor of $\epsilon^{-4/3}$.
>
> > 4. The experimental results are not convincing. For instance, the authors should report the performance of the proposed method when $p= 3$ or $4$.
>
> We greatly appreciate your suggestions for enhancing our experiments. We have now reported the performances of F${}^2$SA-$p$ with $p \in \\{2,3,5,8,10 \\}$ in Figure 1.
>
> ## Reference in the Response
>
> [1] Jeongyeol Kwon, Dohyun Kwon, Stephen Wright, and Robert D. Nowak. _A fully first-order method for stochastic bilevel optimization_. In ICML, 2023.
>
> [2] Lesi Chen, Yaohua Ma, and Jingzhao Zhang. _Near-optimal nonconvex-strongly-convex bilevel optimization with fully first-order oracles_. JMLR, 2025.

---

### Meta-Review · Area_Chair_6DnM · 2026-01-02

**Summary:**

This paper studies first-order stochastic bilevel optimization, where the lower-level problem is strongly convex and the upper-level problem is generally nonconvex. To enable first-order optimization, the authors employ a high-order finite-difference technique to approximate the hypergradient via forward differences of a univariate function. Under appropriate smoothness assumptions, they establish an upper-bound complexity of
$ O(p\kappa^{9+2/p}\epsilon^{-4-2/p}) $, and a lower bound is established to show the tightness of the upper bound under certain conditions.

**Reviewer Concerns:**

All reviewers agree that this paper offers an interesting perspective on first-order bilevel optimization and provides new complexity results under p-th-order smoothness assumptions. The accompanying lower bound demonstrates that the upper bound is nearly tight in certain regimes of p. The reviewers raised several questions. Reviewer Rwzc requested clarification on some technical points and asked for additional experiments; the authors responded convincingly and provided new experimental results. Reviewer MZzi raised several valuable questions regarding specific claims and presentation issues. After carefully reviewing the authors’ responses, I believe these concerns have been addressed satisfactorily, although the reviewer did not provide further feedback. Reviewer pQay questioned the practical verification of the smoothness assumptions.

I also carefully read the paper myself. In my view, the theoretical contributions are substantial and merit publication. The finite-difference approach for bilevel optimization was originally proposed in [1] and this work significantly extends that line of research to settings with high-order smoothness. While the lower-bound construction is nice for demonstrating the near-tightness of the upper bound, it is somewhat less elegant than the upper bound, as the construction effectively reduces the bilevel problem to a single-level one due to independence between levels. Nevertheless, the paper presents novel and meaningful theoretical results in bilevel optimization, and I therefore vote for acceptance.

[1] Achieving $O(\epsilon^{-1.5})$ Complexity in Hessian/Jacobian-free Stochastic Bilevel Optimization. Yang, Y., Xiao, P., and Ji, K. NeurIPS 2023.

**Reviewer Scores:**

The authors provide convincing responses, and I expect the reviewers who initially gave positive evaluations to maintain their scores. Reviewer Rwzc assigned a relatively low score of 4; however, I did not identify any substantial concerns in their comments, and the issues raised appear to have been adequately addressed by the authors. It is therefore possible that this reviewer may increase the score. Overall, the reviewer feedback is positive toward the paper.

---

### Decision · Program_Chairs · 2026-01-26

Accept (Poster)